



# Warming temperatures are impacting the hydrometeorological regime of Russian rivers in the zone of continuous permafrost

Olga Makarieva[1,2], Nataliia Nesterova[2,3], David A. Post[4], Artem Sherstyukov[5], Lyudmila Lebedeva[1]

[1]Melnikov Permafrost Institute, Merzlotnaya St., 36, Yakutsk, Russia 677010
[2]Saint Petersburg State University, Institute of Earth Sciences, 7/9 Universitetskaya nab, St. Petersburg, Russia 199034
[3]State Hydrological Institute, Department of Experimental Hydrology and Mathematical Modelling of Hydrological Processes, 23 2-ya liniya VO, St. Petersburg, Russia 199053
[4]CSIRO, GPO Box 1700, Canberra, Australia
[5]All-Russian Research Institute of Hydrometeorological Information-World Data Center Obninsk, Kaluga oblast, Russia
*Correspondence to*:  Olga Makarieva (omakarieva@gmail.com)

**Abstract.** Analysis of streamflow data was conducted for 22 hydrological gauges in the Yana and Indigirka River basins with a period of observation ranging from 35 to 79 years up to 2015. These river basins are located completely in the zone of continuous permafrost. The main result is the presence of statistically significant ($p < 0.05$) positive trends in monthly streamflow in the autumn-winter period for most of the gauges. Streamflow increases via break points (post 1981) for 17 of the 22 gauges in September (average trend value for the period of record is 58%, or 9.8 mm), and 15 of 22 in October (61%, or 2.0 mm). In November and December, increases are seen in 9 out of 19 (54%, 0.4 mm) and 6 out of 17 non-freezing rivers (95%, 0.15 mm) respectively. Average annual air temperature increases at all 13 meteorological stations of the region by 1.1-3.1 °C over the course of the period 1966-2015. Despite this, the active layer thickness trends are contradictory: it shows an increase of 45 cm, decrease of 77 cm, and no significant trend at the three stations with available soil temperature data. Precipitation decreases in late winter by up to 15 mm over the period of record. Additionally, about 10 mm of precipitation which used to fall as snow at the beginning of winter now falls as rain. Despite the decrease in winter precipitation, no decrease of streamflow is observed during the spring freshet in May and June in the last 50 years (from 1966); moreover, 5 gauges show an increase of 86% or 12.2 mm in spring flood via an abrupt change in 1987-1993. Changes in spring freshet start date were identified for 10 gauges. The trend value varies from 4 to 10 days earlier in May over the period of record. We conclude that warmer temperatures due to climate change are impacting the hydrological regime of these rivers via changes in precipitation type (rain replacing snow). Other factors such as melting of permafrost, glaciers and, aufeises, or changes in groundwater conditions are likely to contribute to these changes in streamflow, but no direct observation of these changes are available. Overall, these changes are likely to have a significant impact on the ecology of the zone of continuous permafrost. Increasing freshwater fluxes to the Arctic Ocean could also impact the Arctic thermohaline circulation. Hydrometeorological data used in this study is combined in a single archive and available at https://doi.pangaea.de/10.1594/PANGAEA.892775.

**Keywords.** Yana and Indigirka Rivers, climate change, warming trend, autumn-winter streamflow increase, precipitation, permafrost, spring freshet



## 1 Introduction

Numerous studies have shown that river streamflow in Northern Eurasia and North America is increasing (Holland et al, 2007; White et al., 2007; Shiklomanov and Lammers, 2009, 2013; Rawlins et al., 2010). Most of them are focused exclusively on the "Big 6" Arctic rivers – the Ob', Yenisey, Lena, Mackenzie, Yukon, and Kolyma (Peterson et al. 2002; Rawlins et al., 2009; Holmes et al., 2013; Rood et al. 2017). Reported streamflow changes are not homogeneous in terms of different runoff characteristics and time scales. While basin-averaged annual and seasonal discharges of the Lena River basin for the period from 1936 to 1995 were increasing (Rawlins et al., 2009), spring peak discharges show no significant changes within the same time frame (Shiklomanov et al., 2007).

Large Arctic river basins are characterized by a great variety of climatic, landscape and permafrost conditions and the mechanisms of the observed changes could hardly be understood on the large scale of the "Big 6". Although climate model simulations project increased net precipitation over the pan-Arctic watershed, this is not always supported by ground meteorological data analysis (Rawlins et al., 2010). Runoff change does not necessarily coincide with precipitation and potential evaporation changes (Gao et al., 2016) but usually agree with increase of air and soil temperature. In some cases there is an opposite change direction in runoff from that in precipitation (Karlsson et al., 2015).

River runoff change estimates in North-Eastern Siberia are limited and contradictory. Magritsky et al. (2013) reported an increase in the total runoff of Yana and Indigirka during the period 1976-2006 by 1.5-3% compared to the period before 1976 and noted that the runoff of these rivers increased in summer and autumn by 20-25% and did not change in winter. According to Georgievsky (2016), on the contrary, there is an increase of Yana and Indigirka rivers winter runoff by 40% over the period 1978-2012 compared with the period 1946-1977, and an increase in the spring flood. Majhi and Daqing (2011) concluded that the Yana River monthly flow rises at the Jubileynaya gauge (224,000 km$^2$) for the period of 1972-1999 for June, August, September, October and April, while May, July and March monthly flow have decreased. Bring and Destouni (2014) reported an absence of any significant changes of the Yana and Indigirka streamflow accompanied by a decrease in precipitation of up to 38% for the period 1991–2002 compared to 1961–1991.

Bring et al. (2016) projected long-term streamflow increases over the Yana and Indigirka rivers by 50% during the 21st century by combining information from model runs of multiple scenarios. These estimates agree with other existing projections (Kuzin and Lapteva, 2015; Shkolnik et al., 2017).

Compared to large rivers, the flow of small and medium rivers in cold regions has been studied much less. Tananaev et al. (2016) found that thirty small and medium-sized rivers out of 100 in the Lena River basin showed trends in mean annual flow. Significant changes have been recently reported for small and middle-sized rivers of northwestern Canada (Spence et al., 2015), Finland (Ashraf et al., 2017), Alaska (Stuefer et al., 2017) and Canadian High Arctic (Lamoureux and Lafrenière, 2017) and attributed to climate change and permafrost disturbances. Further analysis of small river basins could reveal the mechanisms behind ongoing changes as at larger scales, attributing changes can be problematic.

Although air temperature rise and associated permafrost degradation have been reported for the whole Arctic and subarctic territories (Nelson, 2003; Frauenfeld et al., 2004), local anomalies could differ due to specific environmental settings. Three periods with different air temperature tendencies were reported for continental regions




of North-Eastern Siberia: warming period 1916-1945, cooling period 1946-1975 and modern warming period that started in 1976 (Kirillina, 2013). Analysis of Kirillina (2013) precipitation in the region shows positive trend with the exception of the Oymyakon meteorological station.

Permafrost temperature in Russia has been increasing and the active layer has been deepening for the last 20-30 years (Romanovsky et al., 2010; Sherstyukov and Sherstyukov, 2015) but reported changes are spatially heterogeneous and do not necessarily follow air temperature trends. Permafrost degradation could cause greater connectivity between surface and subsurface water (Walvoord and Kurylyk, 2016), talik development (Yoshikawa and Hinzman, 2003; Smith et al., 2005; Jepsen et al., 2013) and other complex hydrological consequences (Quinton et al., 2011; Connon et al., 2014).

Existing assessment of maximum and minimum flow across the cold regions is contradictory. Minimum flow decrease is reported for the Tibetan Plateau (Gao et al., 2016) and Finland (Ashraf et al., 2017) but an increase is observed at Mackenzie River (Yang et al., 2015), at the Lena river and its tributaries (Tananaev et al., 2016) and in most of the Arctic (Rennermalm and Wood, 2010). Shiklomanov et al. (2007) found no widespread significant change in spring maximum discharge, except for the Lena River, among the 139 gauging stations in the Russian Arctic. Tananaev et al. (2016) reported trends in maximum discharge only in nine time series (negative at three and positive at six gauges) out of the 105 in the Lena River basin. The Ob' and Lena rivers showed uncorrelated changes in spring peak with those aggregated from small natural watersheds located within these larger river basins (Shiklomanov et al., 2007). There are significant negative trends ($p < 0.1$) over 1950-2001 from aggregated maximum discharge for Kolyma river basin (Shiklomanov et al., 2007). According to recent projections, the annual maximum river discharge could almost double by the mid-21st century in the outlets of major Siberian rivers (Shkolnik et al., 2017).

Estimation of streamflow changes of small and medium-sized rivers in the Yana and Indigirka river basins do not exist. The objective of this research is a quantitative assessment of current changes of hydrometeorological regime in two large arctic river basins – the Yana and Indigirka, with both basins completely located within the continuous permafrost zone and which have long-term runoff observations along the main rivers and their tributaries at smaller scales.

## 2 Study area

### 2.1 General description, relief

The Yana and Indigirka rivers are two of the few large Arctic rivers whose basins are completely located in the zone of continuous permafrost. Mainly the terrain is mountainous. There is a high elevation of the Verkhoyansk (Orulgan, 2 389 m), Cherskiy (Pobeda, 3 003 m) and Suntar-Khayata (Mus-Khaya, 2 959 m) ranges, as well as wide river valleys. The average altitude of the research basins ranges from 320 to 1410 m and the values of the outlet gauges from -1.55 to 833 m respectively. The lower parts of the Yana and Indigirka basin area are presented by the Yano-Indigirskaya Lowland. The average height is above 30-80 m. Some places of the lowland are raised up to more than 500 m.



### 2.2 Climate

The study territory is the region where the Northern Hemisphere's 'pole of cold' is located. The absolute minimum there has reached record levels: down as far as -71°C in Oymyakon and -68°C in Verkhoyansk (Ivanova, 2006). The research region's climate is distinctly continental. Long-time average annual air temperature changes from -16.1°C (Oymyakon, 726 m, 1930-2012) to -13.1°C (Vostochnaya, 1288 m, 1942-2012). Minimum mean monthly temperatures are typically observed in January and can fall as low as -47.1°C (Oymyakon) and -33.8°C (Vostochnaya). Maximum average monthly temperatures are observed in July, averaging 15.7°C and 11.8°C for stations Yurty (590 m, 1957-2012) and Vostochnaya, respectively.

Steady cold weather begins in the first week of October; spring starts in the second part of May or at the beginning of June, when seasonal snow cover starts to melt. Snow accumulation is relatively small and accounts for 25-30 cm depth.

Annual average precipitation at Verkhoyansk meteorological station (137 m, 1966-2012) is about 180 mm, while at Vostochnaya station (1966-2012) it is 280 mm. Most precipitation (over 60%) occurs in summer, with a peak in July of up to 70 mm per month (Vostochnaya, 1966-2012).

### 2.3 Soil, vegetation and landscapes

The research basins are situated in the transitional zone between forest-tundra and coniferous taiga. For high altitude mountain areas above 1900 m a.s.l., goltsy (bald mountain) and small glaciers are typical. Vegetation in this landscape is absent. Broken stone is present in the form of glacial frost-split boulders as well as diluvia soil of the valley slopes with admixed loam material.

Below is the tundra, which is characterized by distribution of a tight and depressed layer of grass and moss with bushes under which there is rock formation with some ice with admixtures. Most of the research areas are covered larch woodland with moss-lichen cover. In river valleys, grass-moss larch forest and swampy sparse growth forest are typical. Soil types at these landscapes are clayey podzol with partially decayed organic material underlain by frozen soil and bedrock.

### 2.4 Permafrost

Permafrost distribution controls the hydrological regime, especially baseflow formation and ratio of maximum to minimum discharge (Niu et al., 2016). In the studied river basins, permafrost thickness can reach over 450 m at watershed divides and up to 180 m in river valleys and depressions. There are highly dynamic cryogenic features such as retrogressive thaw slumps found in the Yana river basin that indicate on-going permafrost degradation processes (Günther et al., 2015). Permafrost temperature at a depth of zero annual amplitude in the studied region typically vary from -3 to -11 °C, the active layer depth – from 0.3 to 2 m (Explanatory note …, 1991).

### 2.5 Hydrological regime and water balance

The hydrological regime is characterized by spring freshet, high summer-autumn rainfall floods and low winter flow. In winter, small and medium-sized rivers freeze thoroughly. Spring freshet starts in May-June (on average 20th of May) and lasts approximately for a month and a half. In summer, deglaciation waters as well as ones from





melting aufeises and snowfields add to rainfalls.

The amount of precipitation at meteorological stations varies from 176 mm/year (ID 24261) to 279 mm/year (ID 24679). At the high altitude area, annual precipitation can reach up to 600 mm or more per year. Average annual precipitation at Suntar-Khayata station (2068 m) in 1957-1964 reached the value of 690 mm with the corrections for wind and wetting losses. Annual precipitation amount at the mountain peaks is estimated to be as high as 800 mm

(Vasiliev, Torgovkin, 2002; Hydrological Yearbook, 1983).

The average annual flow at studied rivers varies from 58 mm (ID 3433, 18.3 km$^2$) to 362 mm (ID 3516, 16.6 km$^2$). The difference by six times at two watersheds of similar area shows the impact of local conditions on runoff formation. Annual values of flow at the outlet gauges of the Indigirka (ID 3871, 305000 km$^2$) and Yana (ID 3861, 224000 km$^2$) Rivers are 166 and 156 mm respectively.

According to observational data (Makarieva et al., 2018a) annual evapotranspiration depends on the type of underlying surface and the distribution of landscapes over the catchment area. On average, it ranges from 90 mm at the goltsy landscape to 140 mm at the sparse growth of larch trees (Lebedeva et al., 2017).

### 2.6 Groundwater and taliks

Since the whole Yana and Indigirka river basins are located in continuous permafrost, groundwater can be found in a seasonally developed active layer, underneath permafrost (supra-permafrost water) and in taliks. Depending on texture, infiltration and other soil properties, the active layer stores and transmits water to rivers in summer and early autumn. Water in the active layer could sustain recession river flow in autumn only before the freezing front reaches the permafrost table. Taliks with thicknesses of several meters typically exist under small and middle-sized rivers

even if they freeze in winter. Through-taliks are typically found along the river channels with water depth exceeding 3-5 m. Rivers in continuous permafrost often lose water to channel taliks during summer and gain in winter (Arzhakova, 2001). Through-taliks are also associated with fractured deposits with depth exceeding the permafrost thickness (Glotov, 2015). Rivers with lengths >800 km and basin area >75 000 km2 typically do not freeze up and flow over the winter. Year-round groundwater springs and winter flow of large rivers suggest suprapermafrost water

contribution.

### 2.7 Aufeises and glaciers

Aufeises (naleds), which form at mountain foothills, as well as in sub-mountain and intermountain depressions, are another distinguishing feature of the region. The research area has about 10,000 aufeises covering an area of more

than 14,000 km$^2$ (Sokolov, 1975).

The aufeis area share for studied river basins averages from 0.4 to 1.3%, reaching 4% in the basins of some rivers (Tolstikhin, 1974). For example, the aufeis area at the Suntar river (ID 3499) reaches 58 km$^2$ (0.8% of the basin) while in the Charky river basin (ID 3478) it is 113 km$^2$ (1.4%).

In winter, the aufeises reduce river stream and underground flow, and in the warm season, melted aufeis waters form

an additional source of river runoff. Most significant flow from melting aufeis is observed in May-June. In most cases, the share of the aufeis component does not exceed 3-7% of annual river runoff (Reedyk et al., 1995; Sokolov, 1975), but in May its proportion may exceed 50% of the total runoff (Sokolov, 1975).

Few glaciers are found in the research area (GLIMS and NSIDC, 2005, updated 2017). The total area of glaciers is



about 2.2 km² at the Yana river and 436 km² (0.14% of the area) of the Indigirka river basin.

At some studied basins, glaciers reach up to 0.38% of the basin area (3488, Indigirka river – Yurty). The glacier runoff may exceed 3.8% of the overall annual runoff and reach 6.1% of runoff in July and August, as, for example, at the basin of the Agayakan river (the Indigirka basin), where glaciers cover over 1.35% of the catchment (USSR surface waters…, 1972).

On the slopes of the Suntar-Khayata and Cherskiy Ranges, perennial snow fields and rock glaciers are widespread

(Lytkin, Galanin, 2016). They, along with the ice of the active layer and summer atmosphere precipitation, may represent a significant source of the local rivers, however in this respect they are poorly studied (Zhizhin et al., 2012; Lytkin, Galanin, 2016).

### 3   Data and methodology

**3.1 Data**

Daily discharge series for 22 hydrological gauge stations of Russian Hydrometeorological network in the Yana and Indigirka river basins were analyzed. Most of the stations have hydrological data until 2014-2015, two of them ceased operation in 2007 and one (Indigirka river at Vorontsovo) in 1996. The median length of observations series

is 62 years with minimum and maximum values reaching 36 and 80 years respectively (Table 1, Fig. 1).

Ten out of 22 examined catchments have areas less than 10 000 km², with five of them being less than 100 km². Maximum catchment area is 305 000 km² (riv. Indigirka – Vorontsovo). Average elevation above sea level of the examined catchments varies from 320 m (Khoptolooh stream – Verkhoyansk) to 1 410 m (riv. Suntar – riv. Sakhariniya mouth). Thus, the study covers not only large, but also small and medium-sized rivers over a broad

range of elevations and landscapes typical for the studied mountain region.

We used daily discharge data for the entire observation period from 1936 up to and including 2014-2015, published in Hydrological Yearbooks (Hydrological Yearbooks, 1936-1980; State Water Cadastre, 1981-2007) and available for the period 2008-2015 on the website of the Automated information data system for state monitoring of water bodies (AIS SMWB) (URL: https://gmvo.skniivh.ru, reference date: 01.03.2018).

Monthly and annual flow (mm) and spring freshet start dates (counting days from the beginning of the year) were analyzed here. A day was considered as a freshet flood start date if its discharge reached or exceeded 20% of the average discharge value in the studied year. In some cases when daily discharge data was not available, monthly values of flow were adopted from State Water Cadastre (1979, 1987).

Possible errors in flow values shown in the database as reliable are as follows: average annual flow values do not

exceed 10%, monthly flows errors are 10-15% for the open channel period and 20-25% for winter months. The errors of "approximate" flow, placed in the database in parentheses, can exceed the values indicated above by 2-3 times (State water cadastre, 1979, until 1963).

Daily air temperatures and precipitation data series, observed at thirteen weather stations (the elevation range varies from 20 to 1288 m) located in the studied basins, over different periods from 1935 (but not later than 1966) to 2015

(some stations to 2012) were reduced to average monthly values series (Table 2).

Monthly soil temperatures under natural cover at different depths down to 320 cm from three weather stations over a period of 1966-2015 were also analyzed. Detailed description of soil temperature data sets and their quality control





methods may be found in Sherstyukov (2012a; 2012b). Active layer thickness (ALT), used for the analysis, was estimated with polynomial interpolation of soil temperatures at depth (Sherstyukov, 2009; Streletskiy and Sherstyukov, 2015).

The source of the meteorological and soil temperature data is All-Russian Research Institute of Hydrometeorological Information – World Data Centre (http://meteo.ru/data). Combined monthly hydrometeorological data used in this study is available in Makarieva et al. (2018b).

### 3.2 Methods

In this study, we used the combination of statistical methods for the assessment of trends presence and their values in hydrometeorological data as described by Tananaev et al. (2016). Time series of runoff characteristics (monthly flow, estimated flood starting dates and maximum daily flows) and meteorological elements (monthly and annual values of air temperature and precipitation; soil temperature and ALT) were evaluated for stationarity, in relation to presence of monotonic trends, with Mann-Kendall and Spearman rank-correlation tests, at the significance level of $p < 0.05$ (Mann, 1945; Kendall, 1975; Lehmann, 1975). If both tests proved a trend at the significance level $p < 0.05$, a serial correlation coefficient was tested. With the serial correlation coefficient $r < 0.20$, the trend was considered reliable. In the case of $r \geq 0.20$, to eliminate autocorrelation in the input series «trend-free pre-whitening» procedure (TFPW), described by Yue et al. (2002), was carried out. «Whitened» time-series were repeatedly tested with Mann-Kendall non-parametric test at the significance level $p < 0.05$. The Pettitt's test (Pettitt, 1979) was applied to look for the presence of a change point in time series at $p \leq 0.05$, the Buishand range test (Buishand, 1982) was used to search for numerous discontinuities at the same significance level.

Trend values were estimated with Theil-Sen estimator (Sen, 1968) and are given in the relevant data units along with the percentage change since the beginning of observations. Note that the trends are presented for the entire period of observations, and not for the period after the change point was identified, as there can be multiple trends within the period of observations. In some (specified) cases, the significance level was relaxed to the value $p > 0.05$.

## 4 Results

### 4.1 Air temperature

Annual air temperature increase is statistically significant at all 13 studied stations (Table 3, Fig. S1) with an average cumulative value of about +2.1°C (from +0.16 to +0.46 and average +0.35°C per 10 years) for the historical period of observations (minimum, maximum and mean number of years correspond to 47, 80 and 63).

During the period from April to July, there is a significant air temperature trend at most of the weather stations of the region; average trend values in these months account for +3.0°C, 3.1°C, 1.9°C and 2.3°C correspondingly (Table 3). Positive air trend is observed in March in the mountainous part of the Indigirka River (5 stations with elevation > 500 m). In August, air temperature has increased at three Yana basin stations and two lowland stations in the Indigirka basin accounting on average for 3.4 and 2.4°C increase respectively. In September, air temperature has increased at two stations of the lower part of the Indigirka river basin by 2.3°C on average.

There are no significant trends in October at the Yana basin weather stations but mean positive trend at 5 stations of



the Indigirka basin accounting for 3.7°C in the same month. In November, at 7 stations of 13 in both basins positive trends are statistically significant and reach on average 4.7 °C with a maximum value of 6.3°C at Delyankir station (ID 24691).

Positive winter air temperature trends are significant at 4 weather stations in December, 6 – in January and at 1 station – in February. In March positive trend value is 3.6 °C on average for 7 meteorological stations of the mountainous part of the Indigirka River basin.

Change point analysis was conducted for 9 stations out of 13 with full data series. Almost all changes occur through abrupt shifts rather than monotonic trends. Most change points are attributed to two periods: 1975-1985 and 1986-
1996. The shifts in winter air temperature occurred at the beginning of the millennium (1999-2004).

### 4.2 Precipitation

Monthly precipitation analysis for 13 meteorological stations in the region, located at elevations from 20 to 1 288 m, 1966 – 2015 (some stations until 2012), has shown no evidence of a systematic positive trend (Table 4, Fig. S2).

On the contrary, at 6 weather stations out of 9 in the Indigirka basin, a statistically significant precipitation decrease is observed in January and at 3 stations in February. In the Yana basin, 2 stations have shown decrease of precipitation in December. Average statistically significant decrease in winter period accounted for the range from -3.0 to -6.7 mm per month or -44-92 %.

In lowland areas which are close to the furthest most downstream gauges of the Yana and Indigirka rivers, reliable
negative trends were identified in July (-73%, -23 mm) at Chokurdah station (the Indigirka basin) and in August at Kazachie station (-71%, -26 mm) – the Yana basin. It is unlikely that runoff from these areas significantly impacts the flow at the outlets of the Yana and Indigirka Rivers as most of the runoff is formed in upper mountainous parts of the basins.

Positive trends in summer months are observed at two stations of the Indigirka basin – + 44% (21 mm) at
Vostochnaya station in June and +49% (22 mm) and +49% (15 mm) in June and September at Deputatsky station (Table 4).

The evaluation of cold season precipitation calculated as the sum from October to April was carried out. Two stations in the Yana River basin have shown a statistically significant (p<0.05) decrease of about 40% or 20 mm. Three stations in the Indigirka River basin experienced negative trends of about -28% or 15 mm per season at the
level of significance 0.05<p<0.08. The change points in decreasing tendency for solid precipitation are estimated from 1980 to 1996 with most of them occurring in 1986.

Positive trend in warm period precipitation (May – September) is detected at Oymyakon station, the upstreams of the Indigirka river, with the values of 35% or 54 mm. Warm season precipitation has decreased by almost half (-48% or -60 mm) at the outlet of Indigirka river (Chokurdah station).

### 4.2.1 Rain – snow precipitation ratio

The state and timing of precipitation in May and September (which are the months when the 0°C air temperature threshold is crossed) were analyzed. The results show that meteorological stations in the higher elevations (which mostly are located in the Indigirka river basin) exhibit a shift towards larger amounts of rain rather than snow in
both studied months.



In May, three stations show an upward trend for a larger fraction of rain (p<0.05). Vostochnaya station (ID 24679, 1288 m) exhibits the strongest changes: mean share for the whole period is 0.43, trend value is 0.45. The stations Oymyakon and Agayakan (ID 24684, 24688) have mean value of rain share equal 0.79 and on average increase by 0.16 (Table 5).

In September, the mean share of rain fluctuates from 0.56 at Vostochnaya station (ID 24679, 1288 m) to 0.79 at Agayakan and Oymyakon stations (ID 24684, 24688, 726 and 776 m). The trend value varies from 0.15 to 0.19 of increase in rain share for the whole period of observation at 5 stations (Table 5). In absolute values, the amount of rain increased on average by 60.7%, or 12.2 mm in total during the same period at 6 stations. Pettitt and Buishand tests for change points indicate an abrupt shift of precipitation regime in September during the 1991-1993 period at 320 4 stations.

### 4.3 Soil temperature and ALT

There are three meteorological stations with soil temperature data available for the entire historical period including the beginning of the 21st century for studied territory with total area of 529 000 km$^2$ – two in the Indigirka and one 325 in the Yana river basin.

We analyzed soil temperature at a depth of 80 cm and estimated maximum active layer thickness (Table 6). The analysis of changes in the number of days with positive soil temperature at the depth of 80 cm for 2001-2015 in comparison with the long-term mean value for 1971-2000 was also carried out. Positive soil temperatures at the depth of 80 cm at the described weather stations are observed from July to October.

**Verhoyansk, 137 a.s.l., Yana River basin** Significant positive soil temperature trends at the depth of 80 cm in summer (from May to September), and negative ones in winter (from January to March) have been identified (Table 6, Fig. S3).

On average, soil temperature at studied depth increased by +3.4°C in summer months for the last 50 years. Maximum trend values are observed in July and August and account for +4.1 and 4.4 °C respectively. In winter, the 335 corresponding temperature average dropped by -2.2°C for the same period of observations. Winter trends should be viewed with some caution as there was a significant gap in the observations from 1985 to 1999.

Change point in increasing tendency estimated with Pettitt's test was identified around 1996-1999 in June–September and 2003 in May. Accordingly, ALT (mean = 170 cm) increased from by 45 cm. It has grown in 10 years (1997 to 2007) and is stable for the last 8 years. The number of «warm» days at the depth of 80 cm has increased, on 340 average, by 18 days for the last 15 years in comparison with the long-term mean value for 1971-2000 from 87 to 105 cm.

 **Oymyakon, 726 a.s.l., Indigirka River basin** In the uppermost part of the Indigirka river basin, along with meaningful negative soil temperature trends at the depth of 80 cm in summer, positive trends in winter are observed (Table 6, Fig. S3). In June – September soil temperature at 80 cm depth dropped in average by -2.8°C and increased 345 by +4.8°C in October – April.

The change of soil temperature occurs in an abrupt manner – the shifts in winter temperature are identified in 1977 (October), 1983 (November, April), 1990 (May) and 1994-1995 (December – February). In summer, the shift cannot be estimated due to the data gap from 1990-1999.

Due to the temperature decrease in summer, estimated ALT (mean = 165 cm) dropped by 77 cm in the period of



1966-2015 (-1.5 cm a-1). The number of «warm» days has dropped by 32 days from 103 to 71 in the period 2000-2015 in comparison with the previous 30-years period.

**Ust-Moma, 196 a.s.l., Indigirka River basin** Statistically significant positive trends are identified for the period from May to November and on average account for the increase by +2.1°C in 39 years (1977-2015) (Table 6, Fig. S3). Maximum values of trends are observed in October and November and account for +3.0°C and +5.9°C

respectively. The change points are estimated as 2001-2002 for August – November and 2006 for May – July. The trend in ALT (mean=127 cm) is insignificant. The number of «warm» days has increased, on average, by 39 days in 1977 to 2015 from 57 to 96.

### 4.4 Runoff

Positive statistically significant trends ($p<0.05$) in monthly streamflow were identified during two main periods: in autumn-winter and in the first month of a spring flood – in May for most of the river gauges (Fig. 2, Table 7-8). Most of the time series with significant trends are nonstationary, where changes are attributed to break points. Single and double change points are common for studied streamflow records and are described below.

### 4.4.1 Spring flow (May and June)

In May, a statistically significant increase in streamflow is observed at 12 gauges of the 21 studied ones. Those twelve may be divided into two groups.

The first group (Fig. S4, Group A) contains 7 gauges with basin areas from 8290 to 89600 km$^2$ characterized by a longer continuous series of observations beginning from 1937 to 1956. The shifts of streamflow occur in 1964-1966

and follow a similar behaviour. In most cases, a negative tendency changes to an insignificant positive one. Average trend rates are 79% or 8.9 mm for this group. If the assessment of trend is made for the last 50 years after the change point (1966-2015), no significant trend is detected in May for this group of gauges.

The second group (Fig. S5, Group B) consists of 5 gauges with no significant trend, followed by breakpoints during the period from 1980 to 1999 and positive trends thereafter varying from 5.5 to 25.7 mm with average value of 12.2

mm (or from 64 to 103%, 86% in average). Minimum basin area for this group is 16.6, maximum is 52800 km$^2$.

No trends were identified in May at the Yana and Indigirka river outlet gauges for the periods 1972-2007 (ID 3861, 224000 km$^2$) and 1936-1996 (ID 3871, 305000 km$^2$) respectively.

In June, a statistically significant ($p<0.05$) positive trend in streamflow is observed at 4 gauges of the Yana river (Fig. S6). One gauge exhibits a monotonical increase of flow in June for the last 50-60 years and 3 other gauges

have break points in 1983-1995. Average positive trend value is 40% or 16.4 mm.

The small river in the Indigirka basin (ID 3510 – 644 km2, $p<0.057$) has shown total decreased trend value during the last 70 years with a break point in 1967. The streamflow shifts from high mean with negative tendency to low mean with slightly positive tendency. If the analysis is made for the period 1967-2015, streamflow in June at this gauge does not present any detectable changes. Total negative trend value accounts for -61% or -14.2 mm.

### 4.4.2 Summer-autumn floods (July – September)

In July, no statistically significant changes of streamflow occur in the Indigirka river basin, but in general one may note a negative tendency (Table 8). In the Yana River basin, two nested gauges show opposite tendencies (Table 7).





The streamflow at gauge ID 3474 (8290 km²) monotonically decreased by 38% (26 mm). At the same time gauge ID

3443 (52800 km²) experienced total increase of streamflow by 17 mm or 36% with a breakpoint in 1995.

In August, an increase in streamflow was identified at 7 out of 22 studied gauges with an average increase by 55% or 18 mm. Areas of the catchments, where discharge increases in August, vary from 644 to 89600 km². Two nested gauges (ID 3507 and ID 3489) have a monotonic increasing trend from mid-50s to 2015. The other, including two nested gauges of the Adycha river (ID 3443, ID 3445), exhibit the shift in 1982-1987 (Fig. S6, Fig.2).

In September, positive trends were identified at 17 out of 22 gauging stations (average value is 58% or 9.8 mm); three of those rivers are small ones (catchment areas are less than 100 km²), the others are medium and large rivers (Table 7-8, Fig. S7, Fig. 2). This assessment includes a positive trend identified at the Indigirka river outlet gauge (ID 3871) for the shorter period 1936-1996.

Some river gauges of the Yana River basin (ID 3478, 3479, 3483) have shown monotonic increase, other have

change points mainly in the 1981-1982 period (nested gauges ID 3414 and 3424, 3443 and 3445); one gauge (ID 3430) exhibited the shift of mean in 1994.

In the Indigirka river basin, 7 gauges have a change point in 1992-1993 from which two of them with the length of time series of about 70 years have additional change points in 1965.

The Indigirka river gauge (ID 3871) with the data series interrupted in 1996 has a change point in 1965 as well. One

may assume that this gauge would show positive shift in September around 1993 as the other of its tributaries, if longer observation data were available.

### 4.4.3 Low flow (October – March)

In October, streamflow increases at 15 out of 22 gauges (average trend rate is 61% or 2.0 mm) and in November at

11 out of 17 non-frozen gauges (average value is 54% or 0.4 mm) (Table 7-8, Fig. S7-S8). Most of the changes are abrupt. In October most of the step trends occur in 1987-1993 with several exceptions in 1982 and 2001-2002, in November – 1981-1999, mainly around 1994.

In December, positive trends are found at 6 out of 15 non-frozen gauges (Table 7-8, Fig. S8, Fig. 2). Average positive trend magnitude is 95% or 0.15 mm. Change points occur from 1981 to 1994. A negative monotonic trend

in December is found at gauge 3483 with the magnitude -54% or -0.04 mm.

Decrease of streamflow at the Indigirka river at the Vorontsovo (ID 3871) is identified in December and January for the period 1936-1996 and accounts for 32 and 25 % or -0.20 and -0.08 mm respectively with change points in 1964-1969 (Fig. S8-S9, Fig. 2).

In contrary, at the Yana river at Yubileynaya gauges (ID 3861) streamflow increases by 74% (0.12 mm)

monotonically in December and by step trend in 1981 in January amounting to 77% (0.03 mm) correspondingly. Positive trend is also found at the upstreams of the Yana river (gauge ID 3414) in January with the magnitude by 161% or 0.04 mm in 80 years (change point in 1977) (Fig. S9, Fig. 2).

In February, no trends are found at the Yana river basin where only 5 gauges stay unfrozen at this month of the year (Table 7). In March, streamflow trend at the Yana in Yubileynaya is negative with change point around 1989 when

monthly average value of 0.003 mm declines to zero.

Positive trends are identified at two gauges of the Indigirka river (ID 3488, 3489) and one of its main tributary, the Elgi river (ID 3507) with mean magnitude 74% (0.07 mm), 83% (0.02 mm) and 87% (0.08 mm) in February, March





and April respectively (Table 8, Fig. S9). The change points are 1981-1987 for ID 3489 and ID 3507 gauges and 2002-2004 for 3488 station in those three months.

**4.4.4 Annual flow**

In the Yana river basin, statistically significant changes of annual streamflow are found at the Adycha river tributary. Maximum percentage and net flow changes accounting for +51% or +104 mm during 1960-2015 are observed at the Adycha river in Ust'-Charky (ID 3443) with basin area 51100 km$^2$ with a step change in 1995. The

change point at the nested lower gauge ID 3445 (basin area 89600 km$^2$) occurred in 1987 with a total increase of flow by 42% or 82 mm for the period from 1937 to 2015 (Table 7, Fig. S10).

An increasing annual trend is also found at the gauges (ID 3478, 22.6 km$^2$ and ID 3479, 7570 km$^2$) with an increase by 73% or 79 mm with shift in 1988 and monotonic growth by 48 % or 34 mm correspondingly. Monotonic flow changes are observed at two nested gauges of the Yana River (ID 3414, 45300 km$^2$ and ID 3861, 224000 km$^2$) with

the magnitudes by 22%, 24 mm and 39%, 60 mm correspondingly (Table 7, Fig. S10).

In the mountainous part of the Indigirka river basin, positive step trends in annual streamflow are found starting from 1993 at two gauges in the upstream of the Indigirka river (ID 3488, 3489) and its tributary, the Elgi river (ID 3507), with average magnitude by 27% or 49 mm. An increase in annual streamflow is also observed at the smallest basin from the analyzed set (ID 3516, 16.6 km$^2$) with a magnitude of 31% or 115 mm. Runoff in August and

September have contributed most significantly of all months to the annual streamflow rate changes at these gauging stations (Table 8, Fig. S10).

**4.4.5 Maximum daily streamflow**

The analysis of maximum daily streamflow was carried out for the warm period from May to September. In general,

the patterns of changes of maximum daily discharges replicate the change of monthly streamflow. Main changes in May are observed with breakpoints around 1966 when negative trends reverse into an insignificant positive one. The percentage change in May for 8 gauges averages 69% over the whole period of observations. In September, the break points in terms of maximum discharge occur around 1993 in the Indigirka and between 1976-1981 in the Yana River basin. Average increase of maximum discharge in September reaches up to 55% for 15 gauges.


**4.4.6 Freshet onset dates**

In small rivers, freshet starts in the middle of third week of May (May 11-18); for the large rivers this date shifts to the middle of the fourth week (May 25, on average). Air temperature increase in the last decades has led to

significantly earlier freshet starting dates. Freshet starts 4-8 days earlier than 50-70 years ago (the trends are statistically significant) in 8 rivers with the identified streamflow changes in May, as well as in 3 gauges for which monthly streamflow increase in May is statistically insignificant; two of these gauges have catchment areas smaller than 1 000 km$^2$ (Table 1). In the Indigirka River basin, 3 gauges have change points of freshet onset dates in 1967 and another 3 have monotonical shift. In the Yana River basin 4 gauges have monotonical change to earlier onset, 1

gauge has a change point in 1978 and 2 – in 1995-1997.

**5 Discussion**




### 5.1 Air temperature

Global land-surface air temperature has increased over the period 1979-2012 by 0.25-0.27 °C per 10 years (IPCC, 2014). According to Dzhamalov et al. (2012) the warmest decade in Russia was 1990–2000, while the highest temperature was recorded in 2007 (the temperature anomaly of +2.06 °C), followed by 1995 (the anomaly of +2.04 °C) and 2008 (the anomaly of +1.88 °C).

The Arctic has warmed at more than twice the global rate over the past 50 years. The greatest increase of more than
2 °C since 1960 occurred during the cold season (AMAP, 2017). Data from our study supports these observations. The annual air temperature increase in Yana and Indigirka river basins with average cumulative value about +2.1 °C (1966–2015) and trends from +0.16 to +0.46 °C per 10 years slightly exceeds other observations. Interpolated MAAT trends between 1956 and 1990 in the studied region are from 0.15 to 0.30 °C per 10 years (Romanovsky et al., 2007). Kirillina (2013) reported air temperature increase at the Verhoyansk, Ust'-Moma and Oymyakon
meteorological stations from 1 to 2 °C in warm season and from 1.6 to 1.9 °C in winter season for the period 1941-2010.

The different period of analyzed data could partly explain the difference in estimated trends. As noted by Fedorov et al. (2014) and Kirillina (2013), there were several phases with different air temperature tendencies for continental regions of North-Eastern Siberia in the 20th and 21st centuries. The last warming phase started in the 1960s or
1970s and was preceded by three or four decades of cooling. The lowest trends from 0.16 to 0.30 °C/year were identified at the stations with longest MAAT time series that partly included the cooling phase: Ust-Charky 24371 (1942-2015), 21946 Chokurdah (1939-2015), 24679 Vostochnaya (1942-2015) and 24688 Oymyakon (1935-2015). It suggests that the actual MAAT trend for last 40 years is spatially relatively homogenous for the Yana and Indigirka river basins. Trend values exceeding 0.3 °C/year, slightly outnumber globally reported ones and agree
with other regional estimations of 0.03–0.05 °C/year (Pavlov and Malkova, 2009).

### 5.2 Precipitation

Although precipitation is projected to increase over the pan-Arctic basin, this is not always supported by ground meteorological data. Small and insignificant positive trends from 0.63 mm/year to 5.82 mm/year are reported for the
1951–2008 period for the high latitudes of the Northern hemisphere (60°N to 90°N) by IPCC (2014). Pan-Arctic cold season precipitation (October-May) increased by 3.6 mm per decade (1.5% per decade) over the 1936–2009 (AMAP, 2017; Callaghan et al., 2011c).

The presented analysis for 1966-2015 (to 2012 at some stations) has shown no evidence of a systematic positive trend in annual precipitation and even displays a negative trend of solid precipitation for several stations including
Verhoyansk. Savelieva et al. (2000) reference the decrease of winter precipitation over the territory of Eastern Siberia to the changes in the location and intensity of the Siberian High and Aleutian Low before and after 1970s. At the same time Kononova (2013) reported increase of winter sum for Verhoyansk for 1981-2007.

The complex topography of the Arctic region may cause considerable underestimation of precipitation as meteorological stations tend to be located in low elevation areas (Serreze et al., 2003). Analyzed precipitation data
did not undergo wind correction which may vary from 10% in summer to 80-120% in winter due to the effect of wind on gauge undercatch of snowfall (Yang et al., 2005). The study confirms high uncertainty of spatial pattern of





the precipitation trends in cold regions (Hinzman et al., 2013).

### 5.3 Soil temperature

Permafrost temperatures have risen in many areas of the Arctic (AMAP, 2017). Long-term data on permafrost temperature from boreholes for the Yana and Indigirka basins is extremely scarce. Romanovsky et al. (2010) reported an increase of mean annual ground temperature (MAGT) in the eastern part of Northern Yakutia (including the Yana and Indigirka River basins) up to 1.5°C over the last 20 years at the 15 m depth.

Pavlov and Malkova (2009) reported annual ground temperature linear trends of 0.02-0.03 °C/year for North-515 Eastern Russia. The soil temperature data for the studied region is controversial.

Although air temperature increases in May-July period at Oymyakon station, soil temperature dropped on average by -2.8°C (1966-2015) in summer. Partially it could be the consequence of liquid precipitation increase which increased by up to 53 mm (or 36%) in the warm season (May – September). However, precipitation has been growing monotonically and an abrupt drop of soil temperature occurred between 1987 to 2000.

Identified trends of air, soil temperature and precipitation at Verkhoyansk and Ust'-Moma stations agree with each other. At Verkhoyansk soil temperature increase in May-September follows air temperature upward tendency in April-August with one month delay. Soil temperature drop in winter may be caused by decrease of snow depth (Sherstukov, 2008).

Detected trends from -0.05 to +0.12 °C/year at 80 cm depth in different months for 1966 (1977)-2015 at 525 Verkhoyansk and Ust'-Moma stations show significant scatter comparative to reported trends for the whole North-Eastern regions that vary from 0 to 0.03 °C/year.

In Eastern Siberia and the Russian Far East, ALT generally increased between 1996 and 2007 but has since been more stable (AMAP, 2017), which is confirmed by the temporal patterns of ALT change at Verkhoyansk station.

### 5.4 Streamflow

Streamflow significantly ($p<0.05$) increased at least in at least one month at 19 gauges of the 22 analyzed. Three small rivers (ID 3501 84.4 km2, 1120 m, 98 mm; ID 3480 – 1.2 km upstream of the mouth, 98 km2, 570 m, 81 mm; ID 3433, 18.3 km2, 320 m, 58 mm) do not show any significant changes of any studied streamflow metrics.

Analysis of the runoff data for the Yana and Indigirka river basins has shown statistically significant positive 535 discharge trends in May and the autumn-winter period for the last few decades (accompanied by significant warming). Statistically significant increases are seen in12 out of 22 in May, 17 out of 22 in September, 15 out of 22 in October, 9 out of 19 in November, 6 out of 17 in December, 4 out of 12 in January, 3 out of 8 in February and 3 out of 7 in March. Note that total number of gauges decreases below 22 in the period from November to April because the rivers freeze.

#### 5.4.1 Autumn

September shows the most considerable change of hydrological regime. The increase of streamflow happened at basins of all sizes in more than ¾ of the studied gauges. In September, air temperature increased only at 2 meteorological stations out of 13. Although precipitation increased at 10 stations out of 13 in September, only one 545 trend among them is statistically significant.



In the Indigirka river basin, the increase in streamflow can be attributed to the shift of precipitation type in September. It occurs starting from 1993 and manifests in a considerable increase of rain fraction in comparison with snow at meteorological stations located in the mountains. The fact that the changes of streamflow are observed at the gauges regardless of basin size and in general match the breakpoints of streamflow rise and precipitation ratio shift indicate that it is climate rather than other possible factors (permafrost thaw, increased groundwater connectivity, etc.) which drives those changes. Average magnitude of streamflow trend reaches up to 9.8 mm, which is comparable with the mean absolute increase of rain precipitation of 12.2 mm. In the Yana River basin, one would expect a similar linkage between precipitation and streamflow in autumn, but because of the lack of meteorological stations in higher elevations this cannot be confirmed.

Spence et al. (2011) have shown that the trend towards more autumn rainfall in the northwestern subarctic Canadian Shield with no sign of significant total precipitation increase has been sufficient to cause late season peaks in discharge and higher winter low flows. Recessional curves of early autumn flood events extend into later autumn and winter season. The streamflow changes in October generally repeat the spatial pattern of changes in September but with lower magnitude.

The hypothesis of Berghuijs et al. (2014) stating that a shift from a snow- towards a rain-dominated regime would likely reduce streamflow is contradicted by the results of this study for the rivers in the continuous permafrost zone.

### 5.4.2 Winter

Upward trends of low flow are observed in most of the Arctic (Rennermalm and Wood, 2010). The widely found hypothesis is that increased low flow indicates permafrost degradation, aquifer activation and better connectivity between surface and subsurface water. Increasing active layer thickness, enlarged infiltration and sub-surface water contribution to winter discharge by deeper and longer flow pathways sustain increasing winter streamflow in permafrost environments (Karlsson et al., 2015, Niu et al., 2016; St. Jacques and Sauchyn, 2009; Tananaev et al., 2016).

Change points for low flow were detected in 1980s, 1990s and 2000s. Similar change points are observed at the Lena River basin (Tananaev et al., 2016), where the major shifts in all nonstationary time-series occur in the 1990s, but when the data from adjacent decades are combined, the major changes in minimum discharge occur between 1985 and 1995, and in mean annual daily flow between 1995 and 2005.

The changes in November-December tend towards basins larger 17,000 km$^2$ although many smaller ones do not freeze up until January. Glotova and Glotov (2015) attributes that to the fact that the fraction of groundwater contribution to middle- sized and large basins is higher than to smaller ones in the continuous permafrost.

Markov and Gurevich (2008), Gurevich (2009), Dzamalov and Potehina (2010) have substantiated the hypothesis on a regulatory impact of river ice cover in the regions with long-lasting winter on ground water feeding into rivers. It suggests that in colder winters, with a significant thickness of ice, the total water discharge decreases in small river basins. In less severe winters, there is a decrease in the thickness of river ice and the preservation of higher runoff of the underground streamflow in the river by the end of winter.

Shiklomanov and Lammers (2014) report significant negative linear trends for the outlet gauges of the Lena, Yenisey and Yana Rivers where decreases in maximum ice thickness over 1955–2012 reached up to 73, 46 and 33 cm respectively. 10-days series of river ice depth for the Lena River at Tabaga for the period 1955-2015 were





analyzed based on Mann-Kendall test at the significance level of $p < 0.05$. Negative statistically significant trends were identified in March and April with Theil-Sen estimator. The depletion of river ice depth intensifies from March to April and reaches the rates from 27 to 49 cm, or 20-35% (Copernicus Climate Change Service, project #C3S_422_LOT1_SMHI).  However, Shiklomanov and Lammers (2014) have found that the relationship between annual maximum ice thickness and mean river discharge over November–April has shown no significant

correlation: the highest correlation coefficients were found for the Yenisey ($r = −0.63$) and Lena ($r = −0.54$).

### 5.4.3 Freshet

All changes in May occur in an abrupt manner. At 7 gauges (area >=8290 km$^2$) out of 12, the shift of flow happened around 1966, it was accompanied by two consequent years (1967, 1968) with extraordinarily high streamflow in

May. No significant trends are detected for the period after the change point (1967-2015) alone. The changes of spring freshet start date were identified at 10 gauges with streamflow changes in May. Trend value varies from 4 to 10 earlier days. This agrees with Savelieva et al. (2000) who stated that the final frost in spring has been 12–15 days earlier than the mean value in Yakutia. Smith (2000) estimated the magnitude of time shift of start date of spring ice-cover events for the outlet of the Indigirka river at Vorontsovo (ID 3871) as 8.1 days for the period of

observations 1937-1992.

Another 5 basins have shown the shift of flow in May during the period from 1980 to 1999, in most cases, that shift can be attributed to earlier freshet. Yang et al. (2014, 2015) documented increases in May flows for the Mackenzie and Yukon Rivers, respectively related to earlier melting of snow. Earlier start of freshet agrees with identified significant air temperature increase in May at 9 out of 13 meteorological stations in the region. But in contrast to the

Lena river basin, where strong warming in spring led to an advance of snowmelt season into late May and resulted in a lower daily maximum discharge in June (Yang et al., 2002; Tan, 2011), at the study gauges streamflow have not reduced in June (except ID 3510).

### 5.5 Water balance

#### 5.5.1 Precipitation and streamflow relationship

Positive discharge trends in the Arctic have been established in recent years (McClelland et al., 2006; Peterson et al. 2002, 2006; Shiklomanov and Lammers, 2009). Increasing precipitation has been suggested as the one of the driver of increasing streamflow (Dyurgerov and Carter, 2004; McClelland et al. 2004).

The findings of this study largely contradict this, with an increase of streamflow reported at most of the gauges despite no substantial change in absolute amount of precipitation. We estimate that 10 mm of snow is lost in September as rain draining directly into streamflow. Adding the decrease of precipitation in February-March months, the decrease of snow water equivalent during winter may reach up to 15-20 mm in total. Curiously though, no evident decrease of streamflow is observed during freshet or summer months.

Inconsistent trends in discharge and precipitation have also been reported for other Arctic rivers. Berezovskaya et al. (2004) have shown that the patterns in increasing trends of runoff from Siberian Rivers cannot be resolved from the apparent lack of consistent positive trends in the considered precipitation datasets.

Milliman et al. (2008) suggest that a fluctuating climate in the Yana River watershed precluded delineating a



statistically significant change in either precipitation or discharge, but the Indigirka River basin had a statistically significant decrease of precipitation and non-significant increase in runoff.

### 5.5.2 Evapotranspiration

The issue of increasing streamflow in the absence of increasing precipitation may be explained by decreased evapotranspiration. By increasing the active layer depth and thus potentially lowering the water table, evapotranspiration might decrease, leading to increases in runoff (McClelland et al. 2004). Milliman et al. (2008) proposed that decreased evapotranspiration due to earlier snowmelt and the changes in water storage within the drainage basins could be a potential source of excess streamflow. Based on modelling results Stieglitz et al. (2000) showed that as warming occurs mostly during winter months it does not lead to an increase in evapotranspiration. On the other hand, Rawlins et al. (2010) argue that evaporation is growing. This is supported by reanalysis data. According to model simulation, evapotranspiration has a significant trend of 0.11 mm $yr^{-2}$ (VIC model, 1950 – 1999), annual total evapotranspiration from LSMs model also shows positive trend (0.40 mm $yr^{-2}$) (Rawlins et al., 2010). A longer growing season, observed as a result of climate warming in the 20th century, is likely to result in continued upward trends in evapotranspiration (Huntington 2004).

## 5.6 Cryosphere

### 5.6.1 Thawing permafrost

Subsurface ice melting along with air temperature increase, widespread over the studied area (Brown et al., 1998; Yang et al., 2002), can contribute to the streamflow increase (USSR surface waters resources, 1972; Frey and Mclelland, 2009). Though McClelland et al. (2004) argue that if water released from thawing permafrost was making a significant contribution to the observed increase in annual discharge from Eurasian Arctic rivers, one might expect that watersheds with the most permafrost cover would show the largest increase in runoff, but no such pattern is apparent.

An analysis using satellite based gravimetry at the Lena basin suggest that the storage of water in these areas has increased over the past decades (Velicogna et al., 2012; Muskett and Romanovsky; 2009), but the causes of this phenomenon are not clear (Velicogna et al., 2012). One reason may be the permafrost degradation, because warming in the active layer will increase the flow of groundwater and, consequently, the discharge of groundwater into the rivers will be increased too (Ge et al., 2011; Walvoord et al., 2012; Michelle et al., 2016; Lamontagne-Hallé, 2018). However, this phenomenon only holds if sufficient water is available to replenish the increased discharge. Otherwise, there will be an overall lowering of the water table in the recharge portion of the catchment (Ge et al., 2011). Michelle et al. (2016) suggest that permafrost loss is more likely to contribute to baseflow increases in discontinuous permafrost than in continuous permafrost, where permafrost tends to be cold and thick. The results of this study have shown that baseflow is increasing in the zone of continuous permafrost as well.

If the permafrost is degraded, talik zones can be relieved. Most groundwater discharge occurs through areas overlying open taliks so they play the important hydrogeological role in accommodating preferential pathways acting to either recharge the deeper regional aquifer from the supra-permafrost system or facilitating discharge from the deeper aquifer (Bense et al., 2011; Walvoord et al., 2012; Lamontagne-Hallé, 2018). Permafrost thaw can





also generate rapid landscape changes. An example may be thermokarsting and plateau subsidence (Quinton et al., 2011) that in turn influence surface water storage, routing, and runoff (Connon et al., 2014; Michelle et al., 2016).

**665 5.6.2 Glaciers, rock glaciers and aufeises**

Another possible contribution to streamflow increase could be the meltwater from glaciers, at least in the Indigirka river, where glaciers take up to 0.12% of the basin area. The glaciers of the region melt more intensively in July and August so their input to streamflow would be intuitively expected during those months. In the headwaters of the Indigirka river (ID 3488, 51100 km²) the glaciers share is the highest among studied rivers and amounts up to 0.30%, however no significant trend of streamflow is observed in July and August. The next downstream gauge of the Indigirka River (ID 3489, F = 83500 km²) exhibits monotonical increase of streamflow in August from 1944 to 2015 which is mainly explained by increasing streamflow of its tributary, the Elgi River (ID 3507) where no glaciers are found.

However, Liljedahl et al. (2017) have found that in Jarvis Creek (634 km²), (a subbasin of the Tanana and Yukon Rivers), the excess discharge sourced from mountain glaciers has not only increased headwater streamflow, aquifer recharge, and storage but has also increased aquifer capacity – all with the final effect of increasing winter discharge in lower gauges. Ananicheva (2014) estimated the losses of glacier ice volumes for the Suntar-Khayata and Chersky Ranges as 1.7 and 1.4 km3 for the period of 1945-2003 and 1970-2003 respectively. 1.7 km³ in 58 years would give an additional inflow of 0.35 mm per year which by order is comparable with the increase of streamflow in the Indigirka River (ID 3489) in November and December together for the same period. The fact that the change points of the increase of streamflow in late autumn – earlier winter occurred about 10-18 years earlier at downstream gauge (ID 3489) in comparison with headwater gauge (ID 3488) does not contradict this hypothesis. But the same pattern of more early change points at downstream gauge is observed at nested gauges of the Adycha river (ID 3443, 3445) where glacier impact would be negligible due to their tiny area.

The melt of rock glaciers and aufeises, which are widespread in the study area, may have a similar effect of additional water input. According to Lytkin and Galanin (2016), 540 rock glaciers are identified within Suntar-Khayata Range with a distribution density of 8.4 objects per 100 km2. The aufeises account for 0.01 – 0.5% of relative basin areas in the studied territory (Shepelev, 2016).

**690 5.6.3 Geotectonic conditions**

A significant positive trend of annual streamflow was identified only for four gauges. Arzhakova (2001) emphasizes that riv. Elgi (ID 3507, the Indigirka river basin) and riv. Adycha (ID 3443, 3445, the Yana river basin) cross tectonic dislocation zones. The studied area is characterized by modern seismotectonic activities (Imaeva et al., 2016). Taliks, which form within faults and excessive jointing zones, supply rivers even in extreme winter conditions (Romanovsky, 1983; Piguzova, 1989). The correlation between streamflow and sub-permafrost waters is also confirmed by Glotov et al. (2011), who assumed that in particular years, the Kolyma river winter runoff losses at cross sections downstream in comparison with cross sections upstream depending on the current extensive extensions and compressions of underflow through talik space during sublittoral seismic activity periods. Savelieva et al. (2000) pointed to redistribution of the water budget between the upper (thaw layer) water and ground water beneath the permafrost. According Savelieva et al. (2000), the increases of temperature and thickness of the



permafrost active layer may enhance the water inputs from the seasonal thaw layer into the ground waters beneath the permafrost through tectonic trenches and lake's talik zone.

### 5.7 Overall impacts to the ecosystem


This study has identified increases in runoff in rivers discharging to the Arctic Ocean. Increasing volumes of freshwater flows to the Arctic Ocean could lead to a significant weakening of the thermohaline circulation, changing of sea stratification and sea ice formation (Arnell, 2005; Lique et al., 2016, Weatherly and Walsh, 1996). Even subtle changes of river streamflow may have large implication on the ocean-climate system (Miller and Russel, 2000).


Approximately 85% of the total terrestrial runoff to the Arctic Ocean is supplied by rivers draining from the Russian Federation (Aagaard and Carmack, 1989), therefore analysis of this database covering the Yana and Indigirka rivers, is particularly timely. Forman and Johnson (1996) have previously identified the volume, timing, and natural variability of Russian river discharge as being a major priority in Arctic science.


In terms of ecological impacts, increases in discharge, velocity, temperature, and concentration of suspended materials exert important effects on primary production and food-web dynamics in rivers and estuaries (Scrimgeour et al., 1994). Flooding associated with increased discharge is a primary control on the exchange of sediment and organic carbon between rivers and Arctic floodplains (Smith and Alsdorf, 1998).

### 6 Conclusions


Analysis of data from the Yana and Indigirka river basins has shown increases in annual air temperature for the region of around 2.0 °C over the last 50 years (1966-2015). Much of this increase has occurred in the late autumn and spring. Precipitation has shown a decrease in the late winter (February-March) and exhibits little change in other seasons. However, the increases in air temperature have led to a shift in the rain-snow ratio, with more rain at the expense of snow in May and September in mountainous areas. In total, snow loss can be estimated at around 20-25 mm, half of which now falls as rain and the other half is the decrease in late winter precipitation. The increase in air temperature is also reflected in soil temperature and the length of the thaw season with increases at two stations and decreases at one other.


Despite the slight decrease in overall precipitation, analysis of daily discharge data for 22 gauging stations in the Yana and Indigirka river basins (1936-2015) has revealed, at most of the stations, statistically significant (p< 0.05) positive trends in monthly streamflow during the autumn-winter period (September through December) and in the spring flood (May-June). Changes in spring flow are also seen through the spring freshet occurring 4 to 10 days earlier over the period of record. Total annual flow has increased significantly in the upstream tributaries of the Yana and Indigirka rivers.


Most of the increases in streamflow are seen via break points, rather than showing a monotonic increase over the entire period of record. The structure of the timing of these changes is very complex; however, in general, the changes occur in the Yana River basin 10 years before the Indigirka River basin. Changes in winter streamflow are seen in the larger river basins (lower-elevation gauges) before they are seen in the smaller river basins. Analysis of monthly data is required in order to identify these changes, as in many cases, changes in annual flow are not significant. Additionally, analysis of other datasets such as precipitation and temperature is required in order to






develop hypotheses regarding the potential causes of these changes in streamflow.

Increases of streamflow in the autumn-winter period are likely mainly caused by a shift in precipitation from snow to rain in September, with consequent increases in streamflow in that month, continuing into October and later months. Decreased depth of river ice and better connection of surface and ground water due to deepening of active

layer and prolonging of freeze-free period may also be causing higher winter flow.

There are no changes seen in spring discharge in the last 50 years (from around 1966) except at several upstream, mountainous gauges in both studied river basins. Curiously, snow losses and the earlier timing of the spring freshet in May do not lead to a decrease of streamflow in June. This additional input of water is likely related to the regional warming trend impacting more intensive melting of the cryosphere elements such as permafrost, glaciers, and

aufeises.

It is difficult to directly attribute changes in streamflow to definite causes and it is almost certainly a consequence of a complicated interplay between different changing water storages and pathways in warming permafrost. The possibility also exists of additional water input via precipitation higher in the mountains, but this hypothesis cannot be verified with current observational data.

The observed changes agree with reported positive trends in arctic river discharge albeit with low confidence (IPCC, 2014) and the importance of local factors in streamflow response to climate change over Siberia (AMAP, 2017). These changes are having large-scale effects on the Arctic ecosystem, impacting the ecology in the zone of continuous permafrost as well as impacting the livelihood of northern peoples. Increased streamflow may also cause changes to the Arctic Ocean through an increased freshwater flux and decreased thermohaline circulation.

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

**Table 1 Characteristics of runoff gauge stations**

| Index | River – gauge | Period | Basin area (km²) | Outlet catchment elevation (m) | Average catchment elevation (m) | Average annual flow (mm) | The average date of the beginning of the runoff (the number in May) |
|---|---|---|---|---|---|---|---|
| 3414 | Yana – Verkhoyansk | 1936-2015 | 45300 | 125 | 740 | 112 | 21 |
| 3424 | Sartang – Bala | 1957-2015 | 16700 | 136 | 700 | 94 | 19 |
| 3430 | Dulgalaakh – Tomtor | 1956-2015 | 23900 | 145 | 930 | 143 | 20 |
| 3433 | Khoptolookh –Verkhoyansk | 1968-2014 (gap in 1987) | 18.3 | 133 | 320 | 58 | 11 |
| 3443 | Adycha – Ust'-Charky | 1960-2015 | 52800 | 259 | 960 | 203 | 19 |
| 3445 | Adycha – Yurdyuk-Kumakh | 1937-2015 | 89600 | 108 | 880 | 192 | 21 |
| 3474 | Charky - 3.5 km upstream of the mouth | 1949-2007 (gap in 1990) | 8290 | 274 | 1030 | 242 | 19 |
| 3478 | No name (Gnus) - 0.2 km upstream of the mouth | 1953-2007 | 22.6 | 279 | 520 | 115 | 15 |
| 3479 | Borulakh – Tomtor | 1956-2014 | 7570 | 175 | 540 | 71 | 18 |
| 3480 | Turagas – 1.2 km upstream of the mouth | 1969-2014 (gap in 2004, 2012) | 98 | 187 | 570 | 81 | 13 |
| 3483 | Bytantay – Asar | 1945-2015 | 40000 | 80 | 750 | 123 | 24 |
| 3861 | Yana - Yubileynaya (Kazachye) | 1972-2007 | 224000 | -1.55 | 728 | 156 | 27 |
| 3488 | Indigirka – Yurty | 1956-2015 | 51100 | 578 | 1330 | 155 | 19 |
| 3489 | Indigirka – Indigirskiy | 1944-2015 | 83500 | 482 | 1250 | 168 | 21 |
| 3499 | Suntar river – riv. Sakharinya mouth | 1956-2015 | 7680 | 828 | 1410 | 189 | 22 |
| 3501 | Sakharinya –stream mouth | 1957-2014 | 84.4 | 833 | 1120 | 98 | 31 |
| 3507 | Elgi – 5.0 km upstream of the river Artyk-Yuryakh mouth | 1946-2015 | 17600 | 594 | 1140 | 210 | 23 |
| 3510 | Artyk-Yuryakh – 3.5 km upstream of the mouth | 1946-2014 | 644 | 591 | 900 | 89 | 18 |
| 3516 | Dunai (Ambar-Yuryuete) – Rempunkt | 1964-2014 | 16.6 | 494 | 1060 | 362 | 11 |
| 3518 | Nera – Ala-Chubuk | 1945-2015 | 22300 | 568 | 1150 | 174 | 21 |
| 3527 | Blizhniy – 0.3 km upstream of the mouth | 1945-2014 | 23 | 578 | 900 | 108 | 18 |
| 3871 | Indigirka - Vorontsovo | 1936-1996 | 305000 | 3.41 | 760 | 166 | 29 |

*Green color indicates the station in the Yana River basin, brown color – in the Indigirka River basin respectively.*




**Table 2 Characteristics of meteorological stations**

| Index | Meteorological station | X | Y | Elevation, m |
|-------|------------------------|-------|--------|--------------|
| 21931 | Kazachye | 70.75 | 136.22 | 20 |
| 24261 | Batagay-Alyta | 67.80 | 130.38 | 494 |
| 24266 | Verkhoyansk | 67.55 | 133.38 | 137 |
| 24371 | Ust-Charky | 66.80 | 136.68 | 273 |
| 24382 | Ust-Moma | 66.45 | 143.23 | 196 |
| 24585 | Nera | 64.55 | 143.12 | 523 |
| 24588 | Yurty | 64.05 | 141.88 | 590 |
| 24679 | Vostochnaya | 63.22 | 139.60 | 1288 |
| 24684 | Agayakan | 63.33 | 141.73 | 776 |
| 24688 | Oymyakon | 63.27 | 143.15 | 726 |
| 24691 | Delyankir | 63.83 | 145.60 | 802 |
| 24076 | Deputatskiy | 69.33 | 139.67 | 275 |
| 21946 | Chokurdah | 70.62 | 147.90 | 53 |

*Green color indicates the station in the Yana River basin, brown color – in the Indigirka River basin respectively.*





**Table 3 Changes of monthly and annual air temperature (value, year of change point)**

| Index | Period | 1 | 2 | 3 | 4 | 5 | 6 | 7 | 8 | 9 | 10 | 11 | 12 | Avg | *CPY |
|---|---|---|---|---|---|---|---|---|---|---|---|---|---|---|---|
| 21931* | 1961-2015 | 2.8 | 0.7 | 2.9 | **4.1** | **3.1** | 2.0 | 1.4 | **3.0** | 1.7 | 1.8 | 3.0 | 2.1 | **2.2** | **0.40** |
| 24261 | 1966-2012 | *3.1 m* | -3.3 | -0.1 | 1.7 | **3.1 m** | 2.3 | 2.2 | 2.4 | -0.2 | 0.5 | 5.6 2000 | -0.1 | **1.4 m** | **0.30** |
| 24266* | 1969-2015 | **5.6** | 2.1 | 0.0 | **2.3** | **3.8** | **1.7** | **3.2** | **2.1** | 0.5 | 0.6 | 1.9 | **4.5** | 2.1 | **0.45** |
| 24371 | 1942-2015 | **3.5 1990** | 0.7 | 1.0 | **3.4 1967** | **3.3 1970** | 0.8 | *1.5 m* | 0.3 | -0.4 | 0.7 | **4.1 1982** | 1.9 | **1.8 1982** | **0.24** |
| 21946 | 1939-2015 | 1.4 | 0.4 | 1.4 | 2.8 | 1.1 | 0.4 | *1.5 1986* | 2.1 2001 | 2.9 1979 | 4.4 1993 | 4.7 1993 | 2.5 | 2.3 1987 | **0.30** |
| 24076 | 1960-2015 | 4.6 1992 | 1.8 | 2.5 | 3.7 2002 | 3.1 2004 | 0.9 | 1.8 | 2.0 1994 | 1.6 m | 1.7 | 3.2 | 3.1 | 2.5 1987 | **0.45** |
| 24382 | 1938-2015 | 5.1 1975 | 3.0 1978 | 4.1 1983 | 4.5 1980 | 3.5 1987 | 1.1 | 2.2 1986 | 1.4 | 1.3 | 4.5 1987 | 6.3 1983 | 5.2 1978 | 3.6 | **0.46** |
| 24585* | 1966-2012 | -1.6 | 0.1 | **4.0** | 1.4 | 1.8 | **2.2** | 1.7 | 0.7 | 0.3 | 1.5 | **4.7** | **2.9** | **1.7** | **0.36** |
| 24588 | 1957-2015 | 2.4 | 0.1 | 3.3 2000 | 1.0 | 1.2 | 1.4 | *1.6 m* | 1.0 | 0 | 2.1 | **3.8 m** | 2.4 | **1.8 1979** | **0.31** |
| 24679* | 1942-2015 | 1.7 | -0.9 | **2.5** | **1.9** | **2.8** | **1.9** | 1.0 | -0.6 | -0.7 | 0.5 | 2.3 | 1.2 | **1.2** | **0.16** |
| 24684 | 1957-2015 | 1.3 | -0.3 | 3.6 1990 | 1.1 | 1.9 | 2.0 1988 | 2.1 1990 | 1.2 | 0.6 | 1.1 | 5.7 1983 | 3.2 | 2.1 1979 | **0.36** |
| 24688 | 1935-2015 | 3.7 1973 | 0.8 | 4.1 1988 | 2.8 1969 | 2.7 1970 | 2.1 1985 | 1.6 1993 | 0.7 | 0.5 | 1.4 | 1.4 | 1.8 | 2.0 1979 | **0.25** |
| 24691 | 1966-2015 | 1.6 | -0.5 | 2.9 1999 | 1.3 | 1.9 | 1.4 | 2.2 1987 | 1.0 | 0.3 | 3.1 1993 | 6.2 1983 | 5.0 1994 | 2.3 1993 | **0.46** |

Green color indicates the station in the Yana River basin, brown color – in the Indigirka River basin respectively.

The cells filled with grey color and bold fonts correspond to statistically significant trends with p<0.10. If any value is bold it has significance p<0.05; if the values is in italics it has significance 0.05<p<0.10. First figure means trend value (°C), second (if available) means the year of change point. Letter "m" is for monotonical trends. Some stations marked with * had many gaps and it was not possible to assess the change point. CPY (°C (10y)$^{-1}$) is for average change of temperature per 10 years. Statistically significant trends values are divided into 4 groups and marked with different colors accordingly: change points 1985 and before – green, 1985-1995 – violet, 1996 and later – yellow. Monotonous trends and where change points were not available due many gaps are in black.





**Table 4 Changes of monthly, seasonal and annual precipitation (mm, %, change point year), 1966-2015**

| Index | Period | 1 | 2 | 3 | 4 | 5 | 6 | 7 | 8 | 9 | 10 | 11 | 12 | Year | Cold (10-4) | Warm |
|---|---|---|---|---|---|---|---|---|---|---|---|---|---|---|---|---|
| 21931 | 1966-1995, 1999-2015 | 2.5 | -1.4 | 3.0 | 0.0 | -0.7 | -10.0 | -12.9 | **-26.0 -71** | -6.3 | -1.5 | 4.5 | 1.1 | -36.2 | | |
| 24261 | 1966-2012 | -1.7 | -1.8 | -1.2 | -4.5 | 5.3 | 8.1 | 17.9 | 6.5 | 2.0 | -4.2 | -0.3 | **-3.5 -67** | 32.6 | | |
| 24266 | 1966-2015 | -2.9 | -2.9 | -1.3 | -3.1 | 2.2 | 5.8 | 3.0 | 5.1 | 7.3 | -3.9 | -1.9 | -5.0 | 9.0 | *-19.3 -36 1979* | |
| 24371 | 1966-2015 | -1.3 | -1.7 | -0.7 | -4.6 | 1.8 | 13.9 | 5.7 | 8.5 | 5.5 | -4.8 | 2.5 | -4.2 -63 1985 | 10.8 | *-23.6 -46 1996* | |
| 24382 | 1966-2015 | -6.7 -92 1986 | -4.8 -71 m | -2.9 | -1.0 | 4.3 | 5.2 | 3.3 | 7.1 | 4.1 | 0.0 | 0.8 | -4.3 | 11.8 | *-16.0 -2 m* | |
| 24585 | 1966-2015 | **-4.2 -58** | -2.6 | 1.2 | -0.8 | -0.2 | 5.5 | -8.9 | 5.5 | 14.6 | 2.7 | 2.9 | -3.0 | 15.5 | | |
| 24588 | 1966-2015 | **-4.3 -68** | -1.5 | 0.0 | -2.8 | 1.7 | -12.1 | -8.8 | 2.8 | -0.2 | -1.7 | -1.7 | -1.9 | -26.6 | *-14.9 -26* | |
| 24679 | 1966-1993, 1996-2015 | **-4.0 -90** | **-3.3 -121** | -2.1 | -0.4 | 1.3 | **20.7 44** | -28.3 | -3.2 | 13.8 | -1.7 | -0.1 | -1.0 | 41.6 | *-13.0 -30* | |
| 24684 | 1966-2015 | -2.4 | -0.6 | 1.8 | 0.3 | -1.6 | 2.1 | -9.0 | -4.5 | 3.9 | 0.5 | 2.7 | -1.1 | 13.7 | | |
| 24688 | 1966-2015 | -4.4 -60 1980 | -3.0 -44 1994 | -1.3 | -0.6 | 0.5 | 7.5 | 0.1 | 9.7 | 8.5 | -1.0 | 3.0 | -4.5 | 40.9 | | 53 34 m |
| 24691 | 1966-2015 | -5.5 -62 1987 | -3.6 | 1.1 | 1.7 | -3.5 | 9.2 | -2.8 | 12.3 | 5.8 | 3.0 | 3.5 | -3.8 | 15.9 | | |
| 24076 | 1966-2012 | -2.0 | -2.6 | 1.6 | -3.8 | 3.1 | **21.8 49** | 9.4 | 23.7 | **15.3 49** | 5.0 | 4.0 | -1.9 | 58.5 | | |
| 21946 | 1966-2012 | -5.3 | -4.8 | 2.8 | 0.0 | -2.1 | -14.4 | **-22.9 -73** | -11.5 | -1.7 | 5.6 | 5.0 | 3.3 | -42.9 | | -48 -60 |

Green color indicates the station in the Yana River basin, brown color – in the Indigirka River basin respectively.
The cells filled with grey color correspond to statistically significant trends with p<0.10. If any value is bold, it has significance p<0.05; if a value is in italics, it has significance 0.05<p<0.10.

First and second figures mean trend value (mm and %), third (if available) is for the year of change point. Letter "m" is for monotonical trends. If there is neither year, nor "m", the Pettitt's test was not carried out due to many gaps in the data. Statistically significant trends values are divided into 4 groups and marked with different colors accordingly: change points 1985 and before – green, 1985-1995 – violet, 1996 and later – yellow. Monotonous trends and where change points were not available due many gaps are in black.




**Table 5 Characteristics of rain and snow share in precipitation regime in May and September, 1966-2012 (the table contains the data of stations with the changes)**

|  | 24679 | 24684 | 24688 | 24076 | 24585 | 24679 | 24684 | 24688 | 24691 |
|---|---|---|---|---|---|---|---|---|---|
|  | May | | | | | September | | | |
| Significance *p* | <0.01 | 0.01 | 0.01 | 0.15 |  | 0.18 | 0.05 | 0.03 | 0.09 |
| Mean share of rain (dimensionless) | 0.43 | 0.78 | 0.79 | 0.63 |  | 0.56 | 0.79 | 0.79 | 0.74 |
| Trend value (dimensionless) | **0.45** | **0.15** | **0.17** | **0.19** |  | **0.18** | **0.15** | **0.15** | **0.18** |
|  |  |  |  |  |  |  |  |  |  |
| Significance *p* | 0.004 | 0.17 | 0.07 | 0.03 | 0.09 | 0.06 | 0.02 | 0.01 | 0.32 |
| Mean amount of rain (mm) | 8.7 | 12.0 | 10.3 | 19.3 | 25.5 | 53.6 | 19.4 | 18.3 | 20.5 |
| Trend value (%) | 79.0 | 39.9 | 62.8 | 68.1 | 61.0 | 57.6 | 60.3 | 77.7 | 39.3 |
| Trend value (mm) | 6.9 | 4.8 | 6.5 | **13.1** | **15.5** | **10.4** | **11.7** | **14.2** | **8.1** |
|  |  |  |  |  |  |  |  |  |  |
| Pettitt's test significance *p* | 0.03 | 0.06 | 0.23 |  |  | 0.32 | 0.14 | 0.12 |  |
| Pettitt's test change year | 1973 | 1971 | 1987 |  |  | 1979 | **1992** | **1993** |  |
| Buishand test change year | 1986 | 1971 | 2006 |  |  | **1991** | **1992** | **1993** | **1993** |




**Table 6 Changes of soil temperature at 80 cm depth and maximum active layer thickness (ALT)**

|  | 1 | 2 | 3 | 4 | 5 | 6 | 7 | 8 | 9 | 10 | 11 | 12 | ALT | D |
|---|---|---|---|---|---|---|---|---|---|---|---|---|---|---|
| | | | | | 24266 Verkhoyansk, 1966-2015 | | | | | | | | | |
| N | 31 | 31 | 33 | 33 | 48 | 50 | 49 | 49 | 50 | 49 | 49 | 34 | | |
| M | -16.8 | -18.9 | -18.6 | -14.3 | -6.6 | -1.4 | 2.1 | 2.9 | 1.3 | -0.5 | -5.8 | -12.4 | | |
| T | *-1.8* | **-2.3** | **-2.4** | 1.2 | **3.3** | **2.5** | **3.8** | **3.7** | **1.7** | 0.2 | 0.6 | -1.2 | **45** | 18 |
| Y | | | | | 2003 | 1999 | 1996 | 1996 | 1998 | | | | | |
| | | | | | 24688 Oymyakon, 1966-2015 | | | | | | | | | |
| N | 47 | 45 | 44 | 46 | 35 | 38 | 40 | 41 | 43 | 48 | 46 | 47 | | |
| M | -16.1 | -17.5 | -17.4 | -13.8 | -5.3 | -1.0 | 2.3 | 3.1 | 1.5 | -0.2 | -4.9 | -11.7 | | |
| T | **7.6** | **6.3** | **5.0** | **2.5** | -1.8 | **-2.3** | **-4.2** | **-3.5** | *-1.0* | **0.7** | **4.5** | **7.3** | **-77** | -32 |
| Y | 1995 | 1995 | 1990 | 1983 | | | | | | 1977 | 1983 | 1994 | | |
| | | | | | 24382 Ust'-Moma, 1977-2015 | | | | | | | | | |
| N | | | | | 34 | 38 | 38 | 38 | 38 | 39 | 38 | | | |
| M | | | | | -7.3 | -2.5 | 0.3 | 1.0 | 0.3 | -1.5 | -6.7 | | | |
| T | | | | | **1.7** | *1.1* | **1.1** | **1.0** | **1.0** | **3.0** | **5.9** | | 6 | 40 |
| Y | | | | | 2006 | 2006 | 2006 | 2001 | 2001 | 2001 | 2002 | | | |

\*N – the number of values in analyzed series; M, °C – mean temperature; T, °C[1] – temperature trend Sen's estimate; Y – change point (year); ALT (cm) – change of maximum active layer thickness; D – change in the number of days with positive soil temperature at the depth of 80 cm, 2001-2015, comparing to the long-term mean value for 1971-2000. The numbers of row T in **bold** font and filled with grey color are significant values of trends at p<0.05 level. Values in italic font marked by have the level of significance 0.05<p<0.10. The numbers in decreased font are not significant trends. Empty cells refer to the series with significant number of gaps where the assessment of change points were not possible.

Statistically significant trends values are divided into 4 groups and marked with different colors accordingly: change points 1985 and before – green, 1985-1995 – violet, 1996 and later – yellow. Monotonous trends and where change points were not available due many gaps are in black.





**Table 7 Changes of monthly and annual streamflow (mm, %, change point year) and freshet onset dates. The Yana River basin**

| ID | Period | Area, km² | Jan | Feb | Mar | Apr | May | Jun | Jul | Aug | Sep | Oct | Nov | Dec | Year | Freshet onset dates |
|---|---|---|---|---|---|---|---|---|---|---|---|---|---|---|---|---|
| 3478 | 1953-2007 | 22.6 | NA | NA | NA | NA | 6.9 | 14.0 | 12.9 | 13.0 50 1981 | 7.1 54 m | 0.12 31 1987 | 0.00 | 0.00 | 79 69 1988 | 5.8 |
| 3479 | 1956-2014 | 7570 | NA | NA | NA | NA | 5.6 | 12.8 64 m | 4.6 | 8.3 54 1987 | 5.5 67 m | 0.1 | 0.0 | NA | 38 | |
| 3474 | 1949-2007 | 8290 | NA | NA | NA | NA | 7.5 54 1964* | -6.7 | -26 -38 m | 8.7 | 8.4 | 4.5 78 1991 | 0.3 | 0.0 | -7 | 10 |
| 3424 | 1957-2015 | 16700 | 0.0 | NA | NA | NA | 2.9 | 7.3 | -5.5 | 0.4 | 5.7 53 1982 | 0.8 54 1987 | 0.1 | 0.0 | 9 | 4.5 m |
| 3430 | 1956-2015 | 23900 | 0.0 | NA | NA | NA | 5.5 69 1999 | 12.6 | -2.2 | 0.4 | 7.6 50 1994 | 2.0 72 1982 | 0.4 77 1981 | 0.1 149 | 20 | 7.1 1987 |
| 3483 | 1945-2015 (1956) | 40000 | 0.0 | 0.0 | NA | NA | 3.6 71 1966* | 3.4 | 1.8 | 9.0 | 5.7 46 m | 1.3 70 2001 | 0.51 59 1998 | -0.04 -54 m | 24 | |
| 3414 | 1936-2015 (1941) | 45300 | 0.04 161 1977 | 0.0 | 0.0 | 0.0 | 4.3 60 1965* | 7.6 | 3.2 | 5.5 | 5.8 46 1982 | 1.0 51 1993 | 0.3 77 1994 | 0.1 126 1994 | 24 22 m | 6.5 1978 |
| 3443 | 1960-2015 | 52800 | 0.0 | 0.0 | NA | NA | 0.0 | 15.5 79 1987 (1990) | 18.6 28 1995 | 17.4 36 1995 | 27.0 62 1986 | 19.0 83 1981 | 3.3 96 1992 | 0.2 44 1994 | 0.0 | 104 51 1995 | 6.8 1995 |
| 3445 | 1937-2015 (1953) | 89600 | 0.0 | 0.0 | 0.0 | 0.0 | 12.4 83 1966* | 17.9 28 1988 | 7.4 | 24.7 63 1982 | 15.7 69 1981 | 2.1 54 1992 | 0.5 54 1987 | 0.1 75 1987 | 82 42 1987 | 4.3 m |
| 3861 | 1972-2007 | 224000 | 0.03 79 1981 | 0.0 | -0.004 -153 1989 | 0.0 | -0.8 | 16.2 40 1983 | 9.4 | 9.2 | 7.8 | 4.4 118 1993 | 0.4 118 1987 | 0.12 75 m 1982 | 60 40 1983 | |

The cells filled with grey color correspond to statistically significant trends with p<0.10. If any value is bold it has significance p<0.05; if the values is in italics it has significance 0.05<p<0.10.

First and second figures mean trend value (mm and %), third (if available) is for the year of change point. Letter "m" is for monotonical trends. Statistically significant trends values are divided into 4 groups and marked with different colors accordingly: change points around 1966 – red, 1970-1985 – green, 1986-1995 – violet, 1996 and later – yellow. Year of change point marked with * indicates that the gauge has long-term series more of than 70 years with change point in about 1966 and no significant trend after that period (last 50 years). In some cases second year of change point is given in brackets, it was estimated with Buishand range test.



**Table 8 Changes of monthly and annual streamflow (mm, %, change point year) and freshet onset dates. The Indigirka River basin**

| ID | Period | Area, km² | Jan | Feb | Mar | Apr | May | Jun | Jul | Aug | Sep | Oct | Nov | Dec | Year | Freshet onset dates |
|----|--------|-----------|-----|-----|-----|-----|-----|-----|-----|-----|-----|-----|-----|-----|------|--------------------|
| 3516 | 1964-2014 | 16.6 | NA | NA | NA | 0.0 | 25.7 78 1989 | 12.4 | -9.1 | 18.3 | 36.8 111 1992 | 1.6 107 1992 | 0.0 | 0.0 | 115 32 1993 | |
| 3527 | 1945-2014 | 23 | NA | NA | NA | NA | 3.9 | -11.1 | 10.9 | 6.4 | 8.8 87 1965 (1993) | 0.0 | NA | NA | 21 | 4.2 m |
| 3510 | 1946-2014 | 644 | NA | NA | NA | NA | 5.2 | -14.2 -61 1967* | 5.3 | 14.0 65 1982 | 6.3 61 1993 | 0.1 | 0.0 | NA | 21 | |
| 3499 | 1956-2015 | 7680 | NA | NA | NA | NA | 7.2 103 2006 | 11.6 | -17.6 | -2.6 | 9.9 49 1993 | 3.3 70 1993 | 0.43 52 1994 | 0.0 | 26 | 5.4 m |
| 3507 | 1946-2015 | 17600 | 0.03 59 1985 | 0.02 91 1985 | 0.02 86 1985 | 0.01 57 1985 | 15.3 106 1964* | 4.0 | 2.9 | 23.4 53 m | 16.5 76 1993 | 2.1 61 1990 | 0.6 80 1988 | 0.1 39 1987 | 81 39 1995 | 7.6 1967 |
| 3518 | 1945-2015 | 22300 | 0.0 | NA | NA | NA | 11.8 90 1966* | -12.3 | -3.2 | 3.9 | 9.1 51 1965 (1993) | 0.9 34 1966* | 0.2 47 1994 | 0.0 | 14 | 3.9 1967 |
| 3488 | 1956-2015 | 51100 | 0.1 | 0.10 24 2004 | 0.11 57 2004 | 0.10 56 2004 | 7.6 97 1980 | 9.9 | -4.4 | 7.6 | 12.5 59 1993 | 2.3 92 2012 | 0.4 33 1993 | 0.0 | 43 | 5.1 m |
| 3489 | 1944-2015 | 83500 | 0.14 69 1986 | 0.10 86 1977 | 0.10 120 1983 | 0.14 159 1981 | 7.4 92 1966* | -0.6 | -6.0 | 13.2 34 m | 11.4 55 1993 | 1.4 34 1993 | 0.4 28 1982 | 0.3 63 1983 | 33 19 1995 | 4.6 1967 |
| 3871 | 1936-1996 | 305000 | -0.08 -25 1964* | -0.01 | 0.01 | 0.01 | 0.2 | 0.3 | -6.3 | 4.3 | 11 49 1965 | 0.7 | -0.1 | -0.2 -32 1969 | 8.1 | |

All designations are the same as in Table 6.





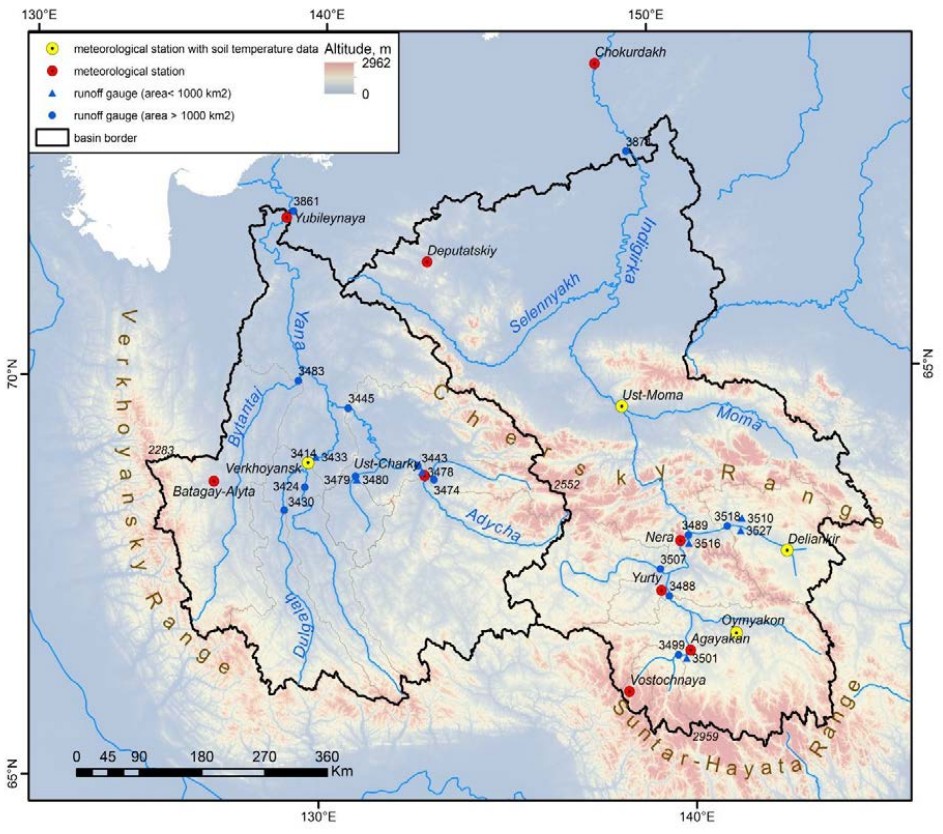


**Figure 1: Meteorological stations and hydrological gauges within the study basins**





**Figure 2: Changes of flow in August, September, October, November, December and annually**
