# Peer review of "Warming temperatures are impacting the hydrometeorological regime of Russian rivers in the zone of continuous permafrost"

_The Cryosphere, 2018_

## Short Comment (SC1) · 21 Sep 2018

This is an important hydro-meteorological analysis for such a data-sparse region, where climate change has huge implication on hydrology and ecosystem. Lots of hydrological data analysis has been done. It will be great if the authors look more at ecohydrological implication. I have not seen any data relating to vegetation. What is vegetation condition there? Does temperature warming increase vegetation coverage during Autumn? Btw, it is good to add a zoom out figure into Figure 1, where once can find where the two basins are.

---

## Referee Comment (RC1) · Anonymous Referee #1 · 7 Nov 2018

General Comments:

This manuscript presents findings after analyzing temperature, precipitation, and streamflow in a previously understudied region. Its content is within the scope of The Cryosphere, and it is well structured. However, it could be made clearer and more convincing by utilizing stronger visuals to communicate results, more thoroughly explaining the novelty of the study, improving sentence structure and grammar, including further discussion, and by using more precise language. I will detail specific examples of changes to make below. The specific comments are not exhaustive of every instance of the above general comments and should not be taken as so.

[Figure]

Specific Comments:

Stronger visuals:

1. In the final paragraph of the introduction, you mention that a study of this kind has never been done. This would be more convincing if you provided a map that shows the study region in context, with sites of other studies labelled. Make it clear that this study is new, and in a region where a study like this is important and required.

2. Section 2.1: Refer to a figure that shows what you describe.

3. Section 2.3: It could be helpful to see the stations plotted over land category.

4. Section 3.1: Histograms or kernel density estimation may be helpful in expressing some of this information. ie: y-axis: frequency, x-axis: catchment size, or elevation above sea level, or length of series, etc.

5. Tables 3-8 are hard to interpret as a reader. Express this information in graphical form, either instead of or in addition to the tables. The green and brown colours are also hard to see as a colour-blind person. You could label the rows instead.

6. Of the figures that are shown in the supplementary information, why were those specific stations chosen?

7. Section 4.2.1: Refer to a figure that shows what you describe.

8. Section 4.4.1: Where do these subclusters exist in space?

9. Section 4.4.5: Is this information displayed somewhere in a table or figure?

Further Discussion:

1. Line 221: Why was twenty percent chosen as the threshold? Is this robust? Do spring rain events ever falsely trigger this threshold?

2. Section 3.2: What assumptions underpin these statistical tests? Can you briefly justify their use here? What are limits to applying these tests?

3. Line 246: "proved" or "suggested"?

4. Line 278: Why was change point analysis not conducted for all thirteen stations?

5. Does the variability of the variables studied change after a change point?

6. Line 505: How can you draw conclusions about trends in winter precipitation if there are uncertainties of over one hundred percent?

7. Line 515: Why is it controversial?

8. Line 517: What is the mechanism that would cause an increase in liquid precipitation to cause soil temperature to drop? I would think that liquid precipitation would warm permafrost through advective heat transport, so please explain to me why I would be wrong.

9. Line 520: You say that the trends agree with each other. Explain the mechanisms that cause you to conclude that they agree.

10. Line 546: Is there a correlation between streamflow to liquid precipitation fraction that you could use to bolster your argument?

11. Section 5.4.3: Are trends in freshet related to elevation of the station? Are changes occurring faster at higher elevations?

12. Section 5.5.2: You should focus this section to discussing your results. You mention a longer growing season, but is agriculture significant in your study region? Is the reanalysis data for the study region? You say that "Rawlins et al. (2010) argue that evaporation is growing"; do you mean evapotranspiration is increasing?

13. Section 5.6.2: In addition to contributing to total streamflow, glaciers tend to reduce the year to year variability of summer streamflow. This is because in hot/dry summers, increased melting partially compensates for reduced rain, while in cool/wet summers, reduced melting partially compensates for increased rain. You could also investigate the year to year variability of July/August total streamflow to further investigate if/where

glaciers are important in these basins.

14. Section 5.7: How does the input from these rivers into the Arctic Ocean compare in magnitude to input from other rivers (such as the Big 6 that you mention earlier)? If the input from your study is much less, then it is harder to link your results to ocean circulation.

15. Line 727: You say that increases in air temperature are reflected in soil temperature and length of the thaw season. Is it? You have soil temperature stations that behave differently than each other and I am not convinced that you can conclude that increases in air temperature are reflected in the soil temperature. I am not saying that the increased temperature isn't reflected in the soil temperature, only that you need to be more clear, explicit, and convincing in your discussion.

16. Line 757: Your study does not investigate how the changes are impacting large-scale features in Arctic ecosystems, and so you cannot conclude that "these changes are having large-scale effects on the Arctic ecosystem". Keep your conclusions within the scope of your results and discussion.

17. Line 758: This is your first mention of how your study relates to the livelihood of northern peoples. Don't introduce new information in your conclusion, and again, keep your conclusions within the scope of your results and discussion.

Other:

1. In the introduction, it is hard to mention the dynamics of major Arctic rivers without mentioning glaciers. Briefly consider the analysis of large-scale basins, including Mackenzie, Yukon, and Ob, for example as presented in "Global-scale hydrological response to future glacier mass loss" by Huss and Hock, 2018

2. Line 85: What are the "other complex hydrological consequences" you refer to?

3. In the final paragraph of the introduction, it would improve clarity if you give more details about what you will do. I.e.: "This study is structured as follows: in Part II, we

[Figure]

will [blank]. In Part III, we will [blank]" etc. Make it clear what your methods will be and how they will achieve the goal you state.

4. Line 119: What is meant by "record"? Is it a regional record? A global record?

5. Line 120: How is "cold" defined?

6. Line 134: How is "small" defined?

7. Line 153: "deglaciation waters" should be "glacier runoff". Also, melt from glaciers, aufeises, and snowfields contribute to streamflow, not rainfall. Be precise with language.

8. Line 175: Does "they" refer to taliks or to rivers?

9. Line 185: A more up to date reference would be better, if possible. I suspect the number of aufeises and their area has changed since the 1970's. Line 186: How does a basin have 4% aufeis if you say that the area share ranges from 0.4 to 1.3 percent?

10. Line 198: Glacier modelling has changed since the 1970's, and so a more up to date reference would be better.

11. Section 2: Are there dams in the region? Are these rivers modified or used by humans in any way (ie: agriculture, municipal water supply)?

12. Section 3.2, and then throughout the study: Trends describe a slope, and in a time series, should be in units of [y-axis unit like mm or C]/time. You need to be more clear about if you are referring to slope of trendline, total difference over period (and how this is calculated), and how percentage change since the beginning of observations is calculated. How do you account for the fact that different stations have different length of observations when you compute the total change over period of observation? Throughout the manuscript you need to be much clearer about what you mean when you refer to trends, total changes, and their units.

13. Line 288: Do you mean per month or per year? It is unclear what this sentence

means.

14. Line 299-301: It sounds like you are talking about a total decrease over the period of observation, but you refer to it as a "decrease" and "trend". Be more clear about what you are talking about.

15. Line 337: What is meant by "Change point in increasing tendency"?

16. Line 387: How can you note a negative tendency if no trends are statistically significant?

17. Line 408: What is a "trend rate"? The units then provided are neither trends nor rates.

18. Line 458: How are "small" and "large" rivers defined?

19. Section 4.4.6: Causal links should be explored in discussion, not in results. This causality needs to be further explored as well, and not just presented as fact.

20. Line 471: What is meant by "highest temperature"?

21. Line 476: What is meant by "average cumulative value"?

22. Line 488: "relatively homogeneous"; relative to what?

23. Line 543: Say the number of stations, not "more than 3/4"

24. Line 544: How do you conclude that precipitation has increased at ten stations if nine trends are insignificant?

25. Line 583: You mention decreases in maximum ice thickness, but this means little without mentioning the mean maximum ice thickness for context.

26. Line 593: What is meant by "changes"? Change in daily streamflow? Be explicit.

27. Line 594: What does "extraordinarily high" mean?

28. Line 601: What does "in most cases" mean when you are talking about five basins?

The phrase "can be attributed" is used without explicitly showing why it can be attributed.

29. Line 671: 0.30% of what? Be clear.

30. Line 685: How is "tiny" defined?

31. Lines 687-688: Why are rock glaciers described in terms of number per 100 km2, while aufeises are described in terms of percentage of basin area?

Sentence Structure, Grammar, and Technical Comments:

1. Line 20-21: Do the 9 out of 19 rivers refer to freezing rivers?

2. Line 52: "agree" should be "agrees" since it refers to "change", not "changes"

3. Line 87: "assessment" should be "assessments" and "is" should be "are"

4. Line 109: "Mainly the terrain is mountainous" should be "The terrain is mainly mountainous"

5. Line 120: Do you mean to say "changes" or "ranges"?

6. Line 207: ", two" should be ", but two"

7. Line 224: "values" or "errors"? Additionally, how do you know the errors of these measurements?

8. Lines 289-291: Restructure this sentence so that its format is the same when describing the trends in July and August.

9. Line 297: The phrase "was carried out" is passive. Use active phrases instead. I.e.: "We evaluated cold season precipitation...". Passive phrasing happens multiple times throughout the manuscript.

10. Line 302: "Positive trend" should be "A positive trend". This type of error occurs multiple times throughout the manuscript.

11. Line 303: "the values" should just be "values".

12. Line 338: "from by" is a typo.

13. Line 340: "for the last fifteen years" or "over the last fifteen years"?

14. Line 342: Have "along with" clause at the end of the sentence.

15. Line 344: "in average" should be "on average"

16. Line 353: "account for the increase by" is unclear. Do you mean "on average, temperature increases by"?

17. Line 366: It is more clear to say "observed at twelve of the twenty-one studied gauges."

18. Line 375: missing units when mentioning the minimum area.

19. Line 379: "monotonical" should be "monotonic"

20. Line 473-474: "the anomaly" should be "anomaly"

21. Line 489: Remove the comma

22. Line 525: "comparative" should be "compared"

23. Line 531: "at least in at least" is a typo.

24. Line 574: "larger" should be "larger than"

25. Line 576: "the continuous permafrost" should be "regions of continuous permafrost", or "the continuous permafrost zone"

26. Line 614: "driver" should be "drivers"

27. Line 661: There is a font change, although this could be an issue of the PDF and not the manuscript.

---

## Referee Comment (RC2) · Anonymous Referee #2 · 3 Dec 2018

This paper examines hydrometerological trends in the Yana and Indigirka rivers basins in the late 20th and early 21st century. Several datasets were used including stream flow data from 22 gauges, air temperature and precipitation from 13 weather stations, and active layer thickness was derived from soil temperature observations at three stations. A range of statistical methods were used, including trend analysis, change point detection, and detection for discontinuities. It is shown that the majority oft the 22 gauging stations have increasing autum-winter stream, many with trends starting after 1981. Air temperature has increased with more than 1 degree between 1966-2015 at all 13 stations. Winter precipitation has decreased, and some has been shifted from snow to rain. However, the drop in winter precipitation has not resulted in decreased

spring freshet. It is concluded that warmer temperature resulting in a shift from snow to rain is driving the hydrological change in the two basins, and suggested that changes in permafrost, glaciers, aufeis, and groundwater conditions may be responsible for streamflow increases.

The central research question: examining hydrometerological trends in Arctic Rivers is an important topic as the Arctic region is undergoing large transformation due to climate change. The authors argue that the streamflow changes in the Yana and Indigirka basins have never been studied before; I do not think this is entirely correct as I am pretty sure pan-Arctic and modeling studies include these basins. Regardless, this study provides a valuable, in-depth, examination of these two basins. The authors are using relevant data and appropriate methods to investigate their research question. It is commendable that the data used in this study is made available for anyone to use in a data repository. The study area description is very comprehensive and gives the reader an excellent understanding of the conditions in these two basins. The discussion provides a rich engagement with previous literature on Arctic hydrometeorological change.

Major comments

1) Some of the most important findings in this study are shown in tables that are very complicated to interpret for a reader, e.g Table 3, 4, 6, 7, and 8. I urge the authors to display the information in figures, which is a more effective way to communicate data to the average reader (see any textbook about data visualization). I do not have the time to analyze these tables as they are currently designed, but I would be happy to provide a complete review of the findings if the authors provide a revised version where the data is shown in a more accessible format.

2) The majority of the figures in the supplementary material are key to study and need to be moved to the main manuscript. Additionally, many figures are only showing a sample. This sample should be motivated, or even better – show all the data.

3) The figure captions can be improved throughout. It should be possible to understand the figures without having to read the manuscript text. Provide more contexts in all captions.

4) I suggest the authors expand their analysis of the spatial pattern of the changes within these two catchments by preparing effective maps. It would help the reader understand if there are spatial clustering and local coherence in trends and changes in various variables.

Minor comments

Study Area Some references are missing, e.g. the sections 2.3, 2.5, and 2.6 lack references about key statements.

Methods 1) Clarify if a separate test of stationary was applied or if stationarity was determined with Mann-Kendall and Spearman rank. 2) Explain why both Mann-Kendall and Spearman rank were used to determine trends 3) Explain the serial correlation better. Why and how was it applied? More details are needed. 4) Use either autocorrelation or serial correlation term to make it easier for the reader to follow along. 5) More context for the Pettitt's test and the Buishand range test would be welcomed.

Results The stations are referred to as numbers in tables 3 and up, but by name in the text (e.g. the section about precipitation). Please choose one or the other, it is too much to ask for the reader to cross-reference with table 1 and 2.

Figure 1: Add an inset map that shows the study area in a larger context (e.g. Siberia)

Figure 2: The symbol size is too small. It is very difficult to see what the changes are. Additionally, the symbology needs to be better explained. Consider removing the background elevation map, which clutters the map and makes it more difficult to interpret.

---

## Author Comment (AC1) · 14 Feb 2019

Reviewer #1 states that the content is within the scope of The Cryosphere and is well structured. They provide numerous suggestions to improve the manuscript by utilizing stronger visuals to communicate results, more thoroughly explaining the novelty of the study, improving sentence structure and grammar, and by using more precise language.

We thank all three reviewers for their positive comments on the manuscript, and their detailed comments to help improve the manuscript. Their numerous, detailed comments will make the manuscript much better overall. We agree that some of the ma-

terial currently presented in tables should also be presented in figures to assist with interpretation and have made those changes. We have also incorporated the vast majority of individual comments as detailed below.

Stronger visuals: Comment: In the final paragraph of the introduction, you mention that a study of this kind has never been done. This would be more convincing if you provided a map that shows the study region in context, with sites of other studies labelled. Make it clear that this study is new, and in a region where a study like this is important and required.

Response: Inset has been added to Figure 1 which shows the study basins in the regional context of the Arctic as a whole. We have modified the text to make it clear that analysis of data for these basins is new. That is, the new part of this study is that we look into the changes of the rivers of different scale in two basins of large rivers which lie fully in the continuous permafrost zone. All other big rivers are situated in different zones (discontinuous, sporadic, no permafrost). We were unable to include a map showing the location of all other studies as we do not have GIS information for these other studies.

Comment: Section 2.1: Refer to a figure that shows what you describe.

Response: Reference to Figure 1 added to Section 2.1. Reference to Figure 2 is added to Section 2.3

Comment: Section 2.3: It could be helpful to see the stations plotted over land category.

Response: Yes, we added a new map showing the landscapes as new Figure 2.

Comment: Section 3.1: Histograms or kernel density estimation may be helpful in expressing some of this information. ie: y-axis: frequency, x-axis: catchment size, or elevation above sea level, or length of series, etc.

Response: As we have only 22 gauges we believe that presenting these data in tabular form in Table 1 is more useful for the reader.

Comment: Tables 3-8 are hard to interpret as a reader. Express this information in graphical form, either instead of or in addition to the tables. The green and brown colours are also hard to see as a colour-blind person. You could label the rows instead.

Response: Extra rows added to Tables 1, 2, 3, and 4 to make it clear which rows correspond to Yana Basin and which to Indigirka Basin in order to remove the issue for colour-blind readers. An additional figure (Figure 1) has been added to the manuscript to represent the changes of monthly and annual stream flow, but the Tables have been retained to provide the reader with the actual numbers as we think that presenting the actual numbers is important.

Comment: Of the figures that are shown in the supplementary information, why were those specific stations chosen?

Response: These stations show the best examples of the behaviour we are describing, and so were therefore included in the paper. The figures were moved to the supplementary section at the request of the Editor on the stage of submitting the paper as they felt there was too much information in the paper itself.

Comment: Section 4.2.1: Refer to a figure that shows what you describe.

Response: This section describes the changes of rain versus snow in May and September. We think the information in Table 5 is sufficient. It is not clear how a figure would make this clearer..

Comment: Section 4.4.1: Where do these subclusters exist in space?

Response: We examined the characteristics of the catchments within the groups and looked into various aspects which could aggregate them in a cluster and could not find any common features, either spatially or with regard to other catchment characteristics, eg size, aspect, latitude, altitude etc.

Comment: Section 4.4.5: Is this information displayed somewhere in a table or figure?

[Figure]

Response: We added an additional table to the supplementary material which provides these data. Further Discussion:

Comment: Line 221: Why was twenty percent chosen as the threshold? Is this robust? Do spring rain events ever falsely trigger this threshold?

Response: The authors derived that 20% threshold by experimenting with different values. To verify that it was working as expected, we checked every gauge for every year visually to confirm the correctness of the date. We also introduced more explanation in the text as the following "There are different ways to determine a freshet start date. For example, Lesack et al. (2013) defined the initiation of freshet discharge based on a threshold value of 3% increase in discharge per day. In this study where non-freezing and freezing rivers were investigated, different approach was considered to be more appropriate. A day was defined as a freshet flood start date if its discharge reached or exceeded 20% of the average discharge value in the studied year. All the stream-flow series were visually checked up for the correctness of such assumption." We also added this publication to the reference list Lesack, L. F. W., P. Marsh, F. E. Hicks, and D. L. Forbes (2013), Timing, duration, and magnitude of peak annual water-levels during ice breakup in the Mackenzie Delta and the role of river discharge, Water Resour. Res., 49, 8234–8249, doi:10.1002/2012WR013198.

Comment: 2. Section 3.2: What assumptions underpin these statistical tests? Can you briefly justify their use here? What are limits to applying these tests?

Response: In this study, we applied the combination of widely used statistical methods for the trend detection and assessment of their values in hydrometeorological data. We really do not think it is necessary to describe those methods in this paper. General guidance with detailed description of different types of tests and their adequate application can be found in Kundzewicz and Robson (2004). We added this reference and the changed the text in the beginning of the Section 3.2. Kundzewicz and Robson, 2004. Change detection in hydrological records—a review of the methodology. Hydrological

[Figure]

Sciences Journal des Sciences Hydrologiques 49: 7–19

Comment: Line 246: "proved" or "suggested"?

Response: Changed to "suggested"

Comment: Line 278: Why was change point analysis not conducted for all thirteen stations?

Response: Because some stations had many gaps in the data which didn't allow for change point test.

Comment: Does the variability of the variables studied change after a change point?

Response: That's a very good question. We didn't study that issue in the current paper, but will look at it further in the future.

Comment: Line 505: How can you draw conclusions about trends in winter precipitation if there are uncertainties of over one hundred percent?

Response: Uncertainties in snow volumes due to wind are well known, we do not have the wind data to correct the precipitation. Therefore, we draw our conclusions on the data we have, giving the reader an understanding of the related uncertainties.

Comment: Line 515: Why is it controversial?

Response: We actually meant multidirectional trends of soil temperature at Oymyakon and Verhoyansk stations. As the issue with Oymyakon data is clarified (see following comment) the data is not controversial. We eliminated this phrase.

Comment: Line 517: What is the mechanism that would cause an increase in liquid precipitation to cause soil temperature to drop? I would think that liquid precipitation would warm permafrost through advective heat transport, so please explain to me why I would be wrong.

Response: The data have shown that the monthly soil temperature at 0.8 m depth

dropped on average by -2.8°Đą (1966-2015) in summer, at the same time air temperature increased and the amount of liquid precipitation increased. We think that the decrease of soil temperature happens due to increase of late summer liquid precipitation which may cause the saturation of clayish soil and the increase of ice content which would take longer to melt in summer. But we have looked at the data more carefully and investigated the temperature at 0.4 m depth. The figure (Figure 2) shows that those radical changes happened in two years – from 1999 to 2001 which seems impossible without any artificial reason. Therefore, we conclude that the data is wrong. We made the following changes in the text Lines 552-557: "Air temperature increases in May-July period at Oymyakon station, soil temperature dropped on average by -2.8°Đą (1966-2015) in summer. Therefore we suggest that the data of Oymyakon station is not reliable as the decrease of soil temperature has happened in abrupt manner which would not be possible without artificial reasons. Additionally, the increase of liquid precipitation would warm permafrost through advective heat transport."

Comment: Line 520: You say that the trends agree with each other. Explain the mechanisms that cause you to conclude that they agree.

Response: The section about soil temperature at Verkhoyansk and Ust'-Moma stations was revised as follows. Lines 560-566: "Identified trends of air, soil temperature and precipitation at Verkhoyansk and Ust'-Moma stations agree with each other. At Verkhoyansk soil temperature increase in May-September follows air temperature upward tendency in April-August with one month delay. Soil temperature drop in winter may be caused by decrease of snow depth (Sherstukov, 2008) due to identified statistically significant decrease of precipitation in cold season from October to April (Table 4). At Ust'-Moma soil temperature increase from May to November could be explained by statistically significant air temperature rise for nine months out of twelve."

Comment: Line 546: Is there a correlation between streamflow to liquid precipitation fraction that you could use to bolster your argument?

Response: We studied the correlation between monthly streamflow and precipitation in September for 4 small watersheds (area < 100 km2) where meteorological stations are located nearby (Figure 3). The correlation coefficient varies from 0.16 to 0.59 for liquid precipitation and from 0.14 to 0.60 for total precipitation (Figure 3-4). We could not explain the reasons of low correlation at gauge 3433 (Figure 3). It is also worth noting that for gauge 3501, the correlation coefficient increases from 0.27 to 0.46 for total and liquid precipitation respectively. We also added more information about that issue in Discussion Section 5.4.1 Autumn

Comment: Section 5.4.3: Are trends in freshet related to elevation of the station? Are changes occurring faster at higher elevations?

Response: No. there are no correlations with either the elevation neither the area of the basin.

Comment: Section 5.5.2: You should focus this section to discussing your results. You mention a longer growing season, but is agriculture significant in your study region? Is the reanalysis data for the study region? You say that "Rawlins et al. (2010) argue that evaporation is growing"; do you mean evapotranspiration is increasing?

Response: We believe that the discussion session is a useful place to put our results in the context of previous findings. This is what we are attempting to do here. The changes of growing season length affect native vegetation as much as agricultural plants. Therefore this issue is within the scope of the discussion. Yes, we clarified the sentences as follows: Lines 681-682: "On the other hand, Rawlins et al. (2010) argue that evapotranspiration is increasing. This is supported by reanalysis data for the pan-Arctic domain."

Comment: Section 5.6.2: In addition to contributing to total streamflow, glaciers tend to reduce the year to year variability of summer streamflow. This is because in hot/dry summers, increased melting partially compensates for reduced rain, while in cool/wet summers, reduced melting partially compensates for increased rain. You could also

investigate the year to year variability of July/August total streamflow to further investigate if/where glaciers are important in these basins.

Response: This is an interesting point, but is beyond the scope of the current paper. We will consider it in future exploration of the data.

Comment: Section 5.7: How does the input from these rivers into the Arctic Ocean compare in magnitude to input from other rivers (such as the Big 6 that you mention earlier)? If the input from your study is much less, then it is harder to link your results to ocean circulation.

Response: Added a comparison of Yana and Indigirka flows to the total inflow to the Arctic Sea to the paper, along with a discussion of potential impacts on the East Siberian Sea and two new references.

Comment: Line 727: You say that increases in air temperature are reflected in soil temperature and length of the thaw season. Is it? You have soil temperature stations that behave differently than each other and I am not convinced that you can conclude that increases in air temperature are reflected in the soil temperature. I am not saying that the increased temperature isn't reflected in the soil temperature, only that you need to be more clear, explicit, and convincing in your discussion.

Response: As the issue with Oymyakon data is clarified (see the reply for the 8 question above) we may conclude that the increases in air temperature are reflected in soil temperature and length of the thaw season. We corrected the text at lines 785-786: "The increase in air temperature is also reflected in soil temperature and the length of the thaw season."

Comment: Line 757: Your study does not investigate how the changes are impacting large scale features in Arctic ecosystems, and so you cannot conclude that "these changes are having large-scale effects on the Arctic ecosystem". Keep your conclusions within the scope of your results and discussion.

Response: Agree. Deleted

Comment: Line 758: This is your first mention of how your study relates to the livelihood of northern peoples. Don't introduce new information in your conclusion, and again, keep your conclusions within the scope of your results and discussion.

Response: Agree. Deleted.

Other: Comment: In the introduction, it is hard to mention the dynamics of major Arctic rivers without mentioning glaciers. Briefly consider the analysis of large-scale basins, including Mackenzie, Yukon, and Ob, for example as presented in "Global-scale hydrological response to future glacier mass loss" by Huss and Hock, 2018.

Response: We added a short consideration of the glacier runoff issue. Lines 55-59: "There are also the studies of glacier retreat and their input into increased stream-flow of the Arctic Rivers (Dyurgerov and Carter, 2004). Bliss et al. (2014) assessing global-scale response of glacier runoff to climate change based scenario modelling have proposed that Canadian and Russian Arctic exhibits steady increases of glacier runoff and affect hydrological regime." Following the comment we also added some more discussion in Section 5.6.2 Glaciers,.. "Huss and Hock (2018) estimated hydro-logical response to current and future glacier mass loss; according to their simulations, annual maximum input of glacier runoff to streamflow in the Indigirka River basin have been passed in the period of 1980-2010 and expected to decline in the future., It will also be accompanied by the change of timing of glacier runoff. In the Indigirka River basin glacier runoff is expected to increase by 20-40% in June and drop by 20% in average in other months (Huss and Hock, 2018)."

Comment: Line 85: What are the "other complex hydrological consequences" you refer to?

Response: We clarified "other complex hydrological consequences". "Permafrost degradation could cause greater connectivity between surface and subsurface water (Walvoord and Kurylyk, 2016), talik development (Yoshikawa and Hinzman, 2003; Smith et al., 2005; Jepsen et al., 2013) and other complex hydrological consequences, such as land-cover changes that can alter basin runoff production in permafrost region (Quinton et al., 2011), and the changes in the wetland-dominated basins that characterize the southern margin of permafrost (Connon et al., 2014)."

Comment: In the final paragraph of the introduction, it would improve clarity if you give more details about what you will do. I.e.: "This study is structured as follows: in Part II, we will [blank]. In Part III, we will [blank]" etc. Make it clear what your methods will be and how they will achieve the goal you state.

Response: Added overview of the paper at the end of the Introduction.

Comment: Line 119: What is meant by "record"? Is it a regional record? A global record?

Response: The temperature reaches record levels for the Northern Hemisphere. Clarified.

Comment: Line 120: How is "cold" defined?

Response: Pole of cold is defined where minimum temperature is observed. It is a widely-used term.

Comment: Line 134: How is "small" defined?

Response: Added minimum and maximum values of glaciers area. ") and small glaciers with minimum and maximum area defined as 0.024 and 5.76 km2 (GLIMS and NSIDC, 2005, updated 2017) are typical"

Comment: Line 153: "deglaciation waters" should be "glacier runoff". Also, melt from glaciers, aufeises, and snowfields contribute to streamflow, not rainfall. Be precise with language.

Response: Corrected

Comment: Line 175: Does "they" refer to taliks or to rivers?

Response: We meant rivers. Corrected to "Taliks with thicknesses of several meters typically exist under small and middle-sized rivers even if the rivers may freeze in winter."

Comment: Line 185: A more up to date reference would be better, if possible. I suspect the number of aufeises and their area has changed since the 1970's.

Response: There is the study of the aufeis changes in the region. It is in review at the moment. The updated assessments and the reference were added. Makarieva, O., Shikhov, A., Nesterova, N., and Ostashov, A.: Aufeis of the Indigirka river basin (Russia): the database from historical data and recent Landsat images, Earth Syst. Sci. Data Discuss., https://doi.org/10.5194/essd-2018-99, in review, 2018.

Comment: Line 186: How does a basin have 4% aufeis if you say that the area share ranges from 0.4 to 1.3 percent?

Response: We wrote that area averages from 0.4 to 1.3. 4% is very high value which is found only for some rivers.

Comment: Line 198: Glacier modelling has changed since the 1970's, and so a more up to date reference would be better.

Response: We carried out a literature search and didn't find any newer assessments of glacier runoff input to streamflow for the studied rivers.

Comment: Section 2: Are there dams in the region? Are these rivers modified or used by humans in any way (ie: agriculture, municipal water supply)?

Response: No, there are no dams in the region. The water can be used for water supply but all the gauges were checked for the information if the streamflow was modified. In this study we used the streamflow which was not modified by any human use.

Comment: Section 3.2, and then throughout the study: Trends describe a slope, and in

a time series, should be in units of [y-axis unit like mm or C]/time. You need to be more clear about if you are referring to slope of trendline, total difference over period (and how this is calculated), and how percentage change since the beginning of observations is calculated. How do you account for the fact that different stations have different length of observations when you compute the total change over period of observation? Throughout the manuscript you need to be much clearer about what you mean when you refer to trends, total changes, and their units.

Response: Because each station has a different length available we are unable to calculate changes over a consistent time period. Most of our results are presented in the form of total change over the whole period. We published the database and give the link to it in the paper, so everyone can explore the data in the way it will be useful to his/her studies. In Section 3.2, we describe how the total changes from the beginning of the observations were estimated: Trend values were estimated with Theil-Sen estimator (Sen, 1968) and are given in the relevant data units along with the percentage change since the beginning of observations. Note that the trends are presented for the entire period of observations, and not for the period after the change point was identified, as there can be multiple trends within the period of observations. In some (specified) cases, the significance level was relaxed to the value ÑĂ>0.05.

Comment: Line 288: Do you mean per month or per year? It is unclear what this sentence means.

Response: Corrected. Average statistically significant decrease in winter months ranges from -3.0 to -6.7 mm per month or -44-92 %.

Comment: Line 299-301: It sounds like you are talking about a total decrease over the period of observation, but you refer to it as a "decrease" and "trend". Be more clear about what you are talking about.

Response: Corrected. Three stations in the Indigirka River basin experienced negative trends at the level of significance 0.05<p<0.08 with reduction of precipitation about -

28% or 15 mm per season.

Comment: Line 337: What is meant by "Change point in increasing tendency"?

Response: Deleted "increasing tendency"

Comment: Line 387: How can you note a negative tendency if no trends are statistically significant?

Response: Deleted "negative tendency"

Comment: Line 408: What is a "trend rate"? The units then provided are neither trends nor rates.

Response: Corrected. In October, streamflow increases at 15 out of 22 gauges (in average by 61% or 2.0 mm) and in November at 11 out of 17 non-frozen gauges (in average by 54% or 0.4 mm) (Table 7-8, Fig. S7-S8).

Comment: Line 458: How are "small" and "large" rivers defined?

Response: We clarified the text. In the rivers with basin area less than 2000 km2, freshet starts in the middle of third week of May (May 11-18); for the larger rivers this date shifts to the middle of the fourth week (May 25, on average).

Comment: Section 4.4.6: Causal links should be explored in discussion, not in results. This causality needs to be further explored as well, and not just presented as fact.

Response: We agree. We eliminated the sentence "Air temperature increase in the last decades has led to significantly earlier freshet starting dates" in this section. Section 5.4.3 contains the discussion of casual links of changes of freshet onset dates.

Comment: Line 471: What is meant by "highest temperature"?

Response: The word anomaly was missing. The sentence is corrected. According to Dzhamalov et al. (2012) the warmest decade in Russia was 1990–2000, while the highest temperature anomaly was recorded in 2007 (temperature anomaly of +2.06

°Đą), followed by 1995 (anomaly of +2.04 °Đą) and 2008 (anomaly of +1.88 °Đą).

Comment: Line 476: What is meant by "average cumulative value"?

Response: Average cumulative value was corrected to average annual value. The annual air temperature increase in Yana and Indigirka river basins with average annual value about +2.1 °Đą (1966–2015) and trends from +0.16 to +0.46 °Đą per 10 years slightly exceeds other observations.

Comment: Line 488: "relatively homogeneous"; relative to what?

Response: It suggests that the actual MAAT trend for last 40 years is spatially homogenous for the Yana and Indigirka river basins. Trend values exceeding 0.3 °Đą/year slightly outnumber globally reported ones and agree with other regional estimations of 0.03–0.05 °Đą/year (Pavlov and Malkova, 2009). Deleted "relatively".

Comment: Line 543: Say the number of stations, not "more than 3/4"

Response: Corrected. "more than 3/4" changed for 17.

Comment: Line 544: How do you conclude that precipitation has increased at ten stations if nine trends are insignificant?

Response: Corrected. In September, air temperature and precipitation increased only at 2 and 1 meteorological stations out of 13 respectively.

Comment: Line 583: You mention decreases in maximum ice thickness, but this means little without mentioning the mean maximum ice thickness for context.

Response: We added average maximum ice thickness for the rivers. "Shiklomanov and Lammers (2014) report significant negative linear trends for the outlet gauges of the Lena, Yenisey and Yana Rivers where decreases in maximum ice thickness over 1955–2012 reached up to 73, 46 and 33 cm respectively. Average values of maximum ice thickness for the same rivers were about 180, 105 and 153 cm respectively for the period 1955-1992 (Vuglinsky, 2000)."

Comment: Line 593: What is meant by "changes"? Change in daily streamflow? Be explicit.

Response: Corrected. The changes of monthly streamflow in May occur in an abrupt manner

Comment: Line 594: What does "extraordinarily high" mean?

Response: "Extraordinarily high" means that out of 22 gauges, at 9 of them monthly streamflow in one of those years was historically highest, at other 6 gauges streamflow was the second highest among the observations with three of them differing less than 5% from historical, other 3 – from 11 to 27%. We added that information as the following: Lines 636-639 "Monthly flow in May 1967 or 1968 exceeded monthly average by 3.8 – 6.5 times. Moreover, at 9 gauges out of 22 it was the highest flow through the whole period of observations. At other 6 gauges streamflow was the second highest among the observations with three of them differing less than 5% from historical maximum, other 3 – from 11 to 27%."

Comment: Line 601: What does "in most cases" mean when you are talking about five basins? The phrase "can be attributed" is used without explicitly showing why it can be attributed.

Response: We corrected for "Another 5 basins have shown the shift of flow in May during the period from 1980 to 1999. In 4 cases of 5 that shift can be attributed to earlier freshet."

Comment: Line 671: 0.30% of what? Be clear.

Response: Corrected. "In the headwaters of the Indigirka river (ID 3488, 51100 km2) the glaciers share is the highest among studied rivers and amounts up to 0.30% of the area share... "

Comment: Line 685: How is "tiny" defined?

Response: We defined "tiny". "But the same pattern of more early change points at downstream gauge is observed at nested gauges of the Adycha river (ID 3443, 3445) where glacier impact would be negligible due to their tiny area (about 1.35 km2 in total)."

Comment: Lines 687-688: Why are rock glaciers described in terms of number per 100 km2, while aufeises are described in terms of percentage of basin area?

Response: Because this is the only information which was found in the literature. Sentence Structure, Grammar, and Technical Comments:

Comment: Line 20-21: Do the 9 out of 19 rivers refer to freezing rivers?

Response: Clarified the sentence. "In November and December, increases are seen in 9 out of 19 rivers which do not freeze in November (54%, 0.4 mm) and 6 out of 17 rivers non-freezing in December (95%, 0.15 mm), respectively."

Comment: Line 52: "agree" should be "agrees" since it refers to "change", not "changes"

Response: Corrected.

Comment: Line 87: "assessment" should be "assessments" and "is" should be "are"

Response: Corrected.

Comment: Line 109: "Mainly the terrain is mountainous" should be "The terrain is mainly mountainous"

Response: Corrected.

Comment: Line 120: Do you mean to say "changes" or "ranges"?

Response: Ranges. Corrected.

Comment: Line 207: ", two" should be ", but two"

Response: Corrected.

Comment: Line 224: "values" or "errors"? Additionally, how do you know the errors of these measurements?

Response: We mean errors. Corrected. Also we have given additional reference on possible errors of streamflow values. "In general the uncertainties associated with discharge determination significantly change from year to year and strongly depend on the computational methods used and frequency of discharge measurements (Shiklomanov et. al, 2006). Possible errors in flow values shown in the database as reliable are as follows for all gauges: average annual flow errors do not exceed 10%, monthly flows errors are 10-15% for the open channel period and 20-25% for winter months. The errors of "approximate" flow, placed in the database in parentheses, can exceed the values indicated above by 2-3 times (State water cadastre, 1979). The errors for large rivers streamflow may be slightly lower: monthly flows errors are 4-12 % for the open channel period and 17 % for winter months in the Lena River basin (Shiklomanov et al., 2006)."

Comment: Lines 289-291: Restructure this sentence so that its format is the same when describing the trends in July and August.

Response: Corrected

Comment: Line 297: The phrase "was carried out" is passive. Use active phrases instead. I.e.: "We evaluated cold season precipitation: : :". Passive phrasing happens multiple times throughout the manuscript.

Response: I am not sure what the reviewer is getting at here. Passive text is common in the scientific literature. No change made.

Comment: Line 302: "Positive trend" should be "A positive trend". This type of error occurs multiple times throughout the manuscript.

Response: Corrected at multiple places in the manuscript.

Comment: Line 303: "the values" should just be "values".

Response: Corrected.

Comment: Line 338: "from by" is a typo.

Response: Corrected.

Comment: Line 340: "for the last fifteen years" or "over the last fifteen years"?

Response: Corrected.

Comment: Line 342: Have "along with" clause at the end of the sentence.

Response: Moved 'along with' to end of sentence as suggested.

Comment: Line 344: "in average" should be "on average"

Response: Corrected.

Comment: Line 353: "account for the increase by" is unclear. Do you mean "on average, temperature increases by"?

Response: Yes. Corrected.

Comment: Line 366: It is more clear to say "observed at twelve of the twenty-one studied gauges."

Response: Corrected.

Comment: Line 375: missing units when mentioning the minimum area.

Response: Corrected.

Comment: Line 379: "monotonical" should be "monotonic"

Response: Corrected.

Comment: Line 473-474: "the anomaly" should be "anomaly"

[Figure]

Response: Corrected.

Comment: Line 489: Remove the comma

Response: Corrected.

Comment: Line 525: "comparative" should be "compared"

Response: Corrected.

Comment: Line 531: "at least in at least" is a typo.

Response: Corrected.

Comment: Line 574: "larger" should be "larger than"

Response: Corrected.

Comment: Line 576: "the continuous permafrost" should be "regions of continuous permafrost", or "the continuous permafrost zone"

Response: Corrected.

Comment: Line 614: "driver" should be "drivers"

Response: Corrected.

Comment: Line 661: There is a font change, although this could be an issue of the PDF and not the manuscript.

Response: Corrected.

Please also note the supplement to this comment:
https://www.the-cryosphere-discuss.net/tc-2018-157/tc-2018-157-AC1-supplement.pdf

[Figure]

**Fig. 1.** Change in monthly streamflow represented as a %, along with the period in which that change occurred. Data are for both Yana and Indigirka river basins and are sorted in order of basin area.

[Figure]

**Fig. 2.** Soil temperature at 0.4 m depth, Oymyakon,1985-2013

| Meteorological station, index | Elevation, m | Gauge index | Average basin elevation, m | Basin area, km$^2$ | Period | Correlation coefficient with precipitation | |
|---|---|---|---|---|---|---|---|
| | | | | | | total | liquid |
| Indigirka River basin | | | | | | | |
| Nera, 24585 | 523 | 3516 | 1060 | 16.6 | 1966-2012 | 0.53 | 0.51 |
| Agayakan, 24684 | 776 | 3501 | 1120 | 84.4 | 1966-2015 | 0.27 | 0.46 |
| Yana River basin | | | | | | | |
| Ust-Charky, 24371 | 273 | 3478 | 520 | 22.6 | 1966-2007 | 0.60 | 0.59 |
| Verkhoyansk, 24266 | 137 | 3433 | 320 | 18.3 | 1967-2015 | 0.14 | 0.16 |

**Fig. 3.** Table 1 Correlation coefficient of monthly streamflow and precipitation in September at small basins

[Figure]

**Fig. 4.** Dependence of streamflow on liquid precipitation in September, Ust-Charky meteoro-
logical station – gauge 3478

**Supplement:**

[revised manuscript text omitted]

---

## Author Comment (AC2) · 14 Feb 2019

Reviewer #2 states that the central research question addresses an important topic and that the paper provides a valuable in-depth examination of the two basins using relevant data and appropriate methods. Their detailed comments focus on improving the communication of results, agreeing with reviewer #1 that the material in some of the tables should be displayed as figures.

We thank all three reviewers for their positive comments on the manuscript, and their detailed comments to help improve the manuscript. Their numerous, detailed comments will make the manuscript much better overall. We agree that some of the ma-

terial currently presented in tables should also be presented in figures to assist with interpretation and have made those changes. We have also incorporated the vast majority of individual comments as detailed below.

Major comments

Comment: Some of the most important findings in this study are shown in tables that are very complicated to interpret for a reader, e.g Table 3, 4, 6, 7, and 8. I urge the authors to display the information in figures, which is a more effective way to communicate data to the average reader (see any textbook about data visualization). I do not have the time to analyze these tables as they are currently designed, but I would be happy to provide a complete review of the findings if the authors provide a revised version where the data is shown in a more accessible format.

Response: An additional figure (Figure 1) was added to the manuscript to represent the main results of the paper – the assessment of changes of monthly and annual stream flow. However, the Tables have been retained to provide the reader with the full set of actual numbers. We agree that our tables need some effort to understand them, but they also have a lot of information about the percentage and absolute value of the change for the whole period and the year of abrupt change. We added the description of general structure of the tables at the beginning of the Section 4 before the description of the results to assist in interpreting them. "The results of trend analysis are presented in the Tables 3-8. The Tables have the same structure and designations. The cells filled with grey color correspond to statistically significant trends with p<0.10. If any value is bold, it has significance p<0.05; if a value is in italics, it has significance 0.05<p<0.10. In Tables 4 (precipitation) and 7-9 (streamflow) each cell with significant trend contains three numbers: 1) the value of total change for the whole period of observations in the characteristic unit (for example, mm) 2) percentage of total change (%); 3) where available – the year of change point or letter "m" for monotonical trend. If there is neither year, nor "m", the Pettitt's test was not carried out due to many gaps in the data. Statistically significant trends values are divided into 4 groups and marked

with different colors accordingly: change points around 1966 – magenta, 1970-1985 – green, 1986-1995 – violet, 1996 and later – yellow. Monotonous trends and where change points were not available due many gaps are in black. For streamflow the year of change point marked with * indicates that the gauge has long-term series more of than 70 years with change point in about 1966 and no significant trend after that period (last 50 years). In some cases second year of change point is given in brackets, it was estimated with Buishand range test. We used the same colors as in the Tables 3-8 in Figure 3 showing the percentage change of monthly and annual streamflow and Figure 4 which presents spatial patterns of change periods."

Comment: The majority of the figures in the supplementary material are key to study and need to be moved to the main manuscript. Additionally, many figures are only showing a sample. This sample should be motivated, or even better – show all the data.

Response: These stations show the best examples of the behaviour we are describing, and so were therefore included in the paper. The figures were moved to the supplementary at the request of the Editor at the stage of submitting the paper.

Comment: The figure captions can be improved throughout. It should be possible to understand the figures without having to read the manuscript text. Provide more contexts in all captions.

Response: We provided more explanations in all captions.

Comment: I suggest the authors expand their analysis of the spatial pattern of the changes within these two catchments by preparing effective maps. It would help the reader understand if there are spatial clustering and local coherence in trends and changes in various variables.

Response: We try to present spatial analysis at Figures 1 and 2. Figure 1 shows the total changes of monthly streamflow in (%) and the periods of changes. The gauges

are sorted by basin area. Figure 2 presents the periods of changes of streamflow in August, September, October, November, December and annually. Red and black colours indicate the presence and absence of trends, respectively.

Minor comments Comment: Study Area. Some references are missing, e.g. the sections 2.3, 2.5, and 2.6 lack references about key statements.

Response: We added additional references in the section 2.3 GLIMS and NSIDC: Global Land Ice Measurements from Space glacier database. Compiled and made available by the international GLIMS community and the National Snow and Ice Data Center, Boulder CO, U.S.A., doi:10.7265/N5V98602, 2005, updated 2017. Fedorov, A.N.; Vasilyev, N.F.; Torgovkin, Y.I.; Shestakova, A.A.; Varlamov, S.P.; Zheleznyak, M.N.; Shepelev, V.V.; Konstantinov, P.Y.; Kalinicheva, S.S.; Basharin, N.I.; Makarov, V.S.; Ugarov, I.S.; Efremov, P.V.; Argunov, R.N.; Egorova, L.S.; Samsonova, V.V.; Shepelev, A.G.; Vasiliev, A.I.; Ivanova, R.N.; Galanin, A.A.; Lytkin, V.M.; Kuzmin, G.P.; Kunitsky, V.V. Permafrost-Landscape Map of the Republic of Sakha (Yakutia) on a Scale 1:1,500,000. Geosciences, 8, 465. 2018. Section 2.5 Grave N., Gavrilova M., Gravis G., Katasonov E., Klyukin N., Koreysha G., Kornilov B., Chistotinov L. The freezing of the earth's surface and glaciation on the ridge Suntar-Hayata (Eastern Yakutia). Nauka, Moscow. 1964 (in Russian) Hydrological Yearbook: Volume 8. Issue. 0-7. The basin of the Laptev and East-Siberian seas to the Kolyma river, Yakutsk Department of Hydrometeorology, Yakutsk, 1936-1980. State water cadastre: Annual data on the regime and resources of surface terrestrial waters. Volume 1. Issue 16. The Lena River basin (middle and lower course), Khatanga, Anabara, Olenka, Yana, Indigirka, Yakutsk Department of Hydrometeorology, Yakutsk, 1981-2007. Section 2.6 Shepelev, V.V.: Suprapermafrost waters in the cryolithozone. Novosibirsk. Geo. 2011, 169 pp (in Russian) Mikhailov, V. M.: Floodplain taliks of North-East of Russia. Novosibirsk. Geo. 2013, 244 pp. (in Russian)

Methods Comment: Clarify if a separate test of stationary was applied or if stationarity was determined with Mann-Kendall and Spearman rank.

Response: The stationarity of the time series was checked with respect to: 1) a monotonous trend (Mann-Kenndall and Spearman) and 2) abrupt changes (Pettitt's and Buishand tests).

Comment: Explain why both Mann-Kendall and Spearman rank were used to determine trends.

Response: In most cases the interpretations of Kendall's tau and Spearman's rank correlation coefficient are very similar. Two tests were selected mostly selected to check and compare the results because no statistical test is perfect even when all test assumptions are met; more than one statistical test is good practice (Kundzewicz and Robson, 2004. Change detection in hydrological records—a review of the methodology. Hydrological Sciences Journal des Sciences Hydrologiques 49: 7–19).

Comment: Explain the serial correlation better. Why and how was it applied? More details are needed.

Response: Serial correlation increases the number of errors of the first kind when checking for the presence of a trend, overestimating the significance of the assessment, and the probability of finding a trend where there is none in reality increases. On the other hand, the presence of a stationary trend overestimates the value of the autocorrelation coefficient. The method proposed in [Yue et al., 2002] and known as trend-free pre-whitening (TFPW) was used to increase the reliability of statistical trend assessment. At the first step, the linear component is subtracted from the time series, the coefficient of which is determined by the Theil-Sen method. In the second step, the time series is decorrelated by subtracting from it the component corresponding to the first-order AR (1) autoregressive process. Then the two series are summarized, after which the values of the rank correlation indicators are determined for the final series.

Comment: Use either autocorrelation or serial correlation term to make it easier for the reader to follow along.

Response: Used serial correlation. Corrected.

Comment: More context for the Pettitt's test and the Buishand range test would be welcomed.

Response: The Pettitt's test and the Buishand range test are widely used to identify change points in series of hydrometeorological data. The Pettit test for a change in the median of a series with the exact time of change unknown is based on ranks, which implies that it is less sensitive to outliers. The Buishand test is used to detect a change in the mean by studying the cumulative deviation from the mean, it assumes a normal distribution of data.

Results Comment: The stations are referred to as numbers in tables 3 and up, but by name in the text (e.g. the section about precipitation). Please choose one or the other, it is too much to ask for the reader to cross-reference with table 1 and 2.

Response: Corrected.

Comment: Figure 1: Add an inset map that shows the study area in a larger context (e.g. Siberia)

Response: Change made as requested (Figure 3).

Please also note the supplement to this comment:
https://www.the-cryosphere-discuss.net/tc-2018-157/tc-2018-157-AC2-supplement.pdf

[revised manuscript text omitted]

---

## Author Comment (AC3) · 14 Feb 2019

Y. Zhang states that this is an important analysis in a data-sparse region and recommends looking at ecohydrological implications more closely, particularly whether warming increases vegetation cover during Autumn. He also suggests adding an inset to Figure 1 to put the basins into regional context. We thank Dr Zhang for valuable suggestions for future studies.

Comment: It will be great if the authors look more at ecohydrological implication. I have not seen any data relating to vegetation. What is vegetation condition there? Does temperature warming increase vegetation coverage during Autumn?

[Figure]

Response: There is a brief description of the vegetation in Section 2.3. Also, we added a Figure 1 which shows the spatial distribution of vegetation. We did not look into vegetation changes with climatic changes yet, but this is a good suggestion. It would be interesting to look at long term trends in vegetation, and we will follow this up. However, changes in monthly temperature occur in July. So it is unlikely to increase vegetation coverage in autumn.

Comment: It would be good to add a zoom out figure into Figure 1, where once can find where the two basins are.

Response: Inset has been added to Figure 2. It shows the location of the basins within the Arctic region as a whole.

[Figure]

**Fig. 1.** Landscape distribution within the study basins according to Permafrost-Landscape Map of the Republic of Sakha (Yakutia) on a Scale 1:1,500,000 (Fedorov et al., 2018)

**Fig. 2.** Meteorological stations and hydrological gauges within the study basins